Brief Communication

# Assessing GPT-4 for cell type annotation in single-cell RNA-seq analysis

Wenpin Hou ®[1] ✉ & Zhicheng Ji ®[2] ✉

Here we demonstrate that the large language model GPT-4 can accurately annotate cell types using marker gene information in single-cell RNA sequencing analysis. When evaluated across hundreds of tissue and cell types, GPT-4 generates cell type annotations exhibiting strong concordance with manual annotations. This capability can considerably reduce the effort and expertise required for cell type annotation. Additionally, we have developed an R software package GPTCelltype for GPT-4's automated cell type annotation.

Cell type annotation is a fundamental step in single-cell RNA sequencing (scRNA-seq) analysis. This process is often laborious and time-consuming, requiring a human expert to compare genes highly expressed in each cell cluster with canonical cell type marker genes. Although automated cell type annotation methods have been developed (Supplementary Table 1), manual annotation using marker genes remains widely used.

Generative pre-trained transformers (GPT), including GPT-3.5 and GPT-4, are large language models designed for language understanding and generation. Recent studies have demonstrated their effectiveness in biomedical contexts[1,2]. In this Brief Communication, we hypothesize that GPT-4 can accurately annotate cell types, transitioning the annotation process from manual to a semi- or even fully automated procedure (Fig. 1a). GPT-4 offers cost-efficiency and seamless integration into existing single-cell analysis pipelines such as Seurat[3], avoiding the need for building additional pipelines and collecting high-quality reference datasets. The vast training data of GPT-4 enables broader applications across various tissues and cell types, and its chatbot nature allows for user-driven annotation refinement (Fig. 1a,b).

We systematically assessed GPT-4's cell type annotation performance across ten datasets[4–12], covering five species and hundreds of tissue and cell types, and including both normal and cancer samples (Supplementary Table 2). GPT-4 was queried using GPTCelltype, a software tool we developed (Methods). For competing methods, we evaluated GPT-3.5, a prior version of GPT-4, and CellMarker2.0[13], SingleR[14] and ScType[15], which are automatic cell type annotation methods that provide references applicable to a large number of tissues (Methods and Supplementary Table 1). Cell type annotations by GPT-4 or competing methods were evaluated based on their agreement with manual annotations provided by the original studies. The degree of agreement

was measured using a numeric score (Methods). Supplementary Table 3 presents an example of evaluating GPT-4 cell type annotations in human prostate tissue, and details of all cell type annotations and their evaluation results are included in Supplementary Table 4.

We first explored different factors that may affect the annotation accuracy of GPT-4 (Fig. 2a and Supplementary Table 5). We found that GPT-4 performs best when using the top ten differential genes, and when differential genes are derived using the two-sided Wilcoxon test. GPT-4 exhibits similar accuracy across various prompt strategies, including a basic prompt strategy, a chain-of-thought[16]-inspired prompt strategy that includes reasoning steps, and a repeated prompt strategy (Methods). In subsequent analyses, both GPT-4 and GPT-3.5 used the basic prompt strategy with the top ten differential genes obtained from Wilcoxon test as inputs for applicable datasets.

GPT-4's annotations fully or partially match manual annotations in over 75% of cell types in most studies and tissues (Fig. 2b), demonstrating its competency in generating expert-comparable cell type annotations. This agreement is particularly high for marker genes from literature searches, with at least 70% fully match rate in most tissues. Though lower for genes identified by differential analysis, the agreement remains high. However, results from datasets published before September 2021 should be interpreted cautiously as they predate GPT-4's training cutoff. GPT-4 performs better for immune cells like granulocytes compared to other cell types (Fig. 2b). It identifies malignant cells in colon and lung cancer datasets but struggles with B lymphoma, potentially due to a lack of distinct gene sets. The identification of malignant cells could benefit from other approaches such as copy number variation[9]. Performance dips slightly in small cell populations comprising no more than ten cells (Fig. 2b), possibly due to the limited available information. GPT-4 annotations fully match manual

[1]Department of Biostatistics, Columbia University Mailman School of Public Health, New York City, NY, USA. [2]Department of Biostatistics and Bioinformatics, Duke University School of Medicine, Durham, NC, USA. ✉e-mail: wh2526@cumc.columbia.edu; zhicheng.ji@duke.edu

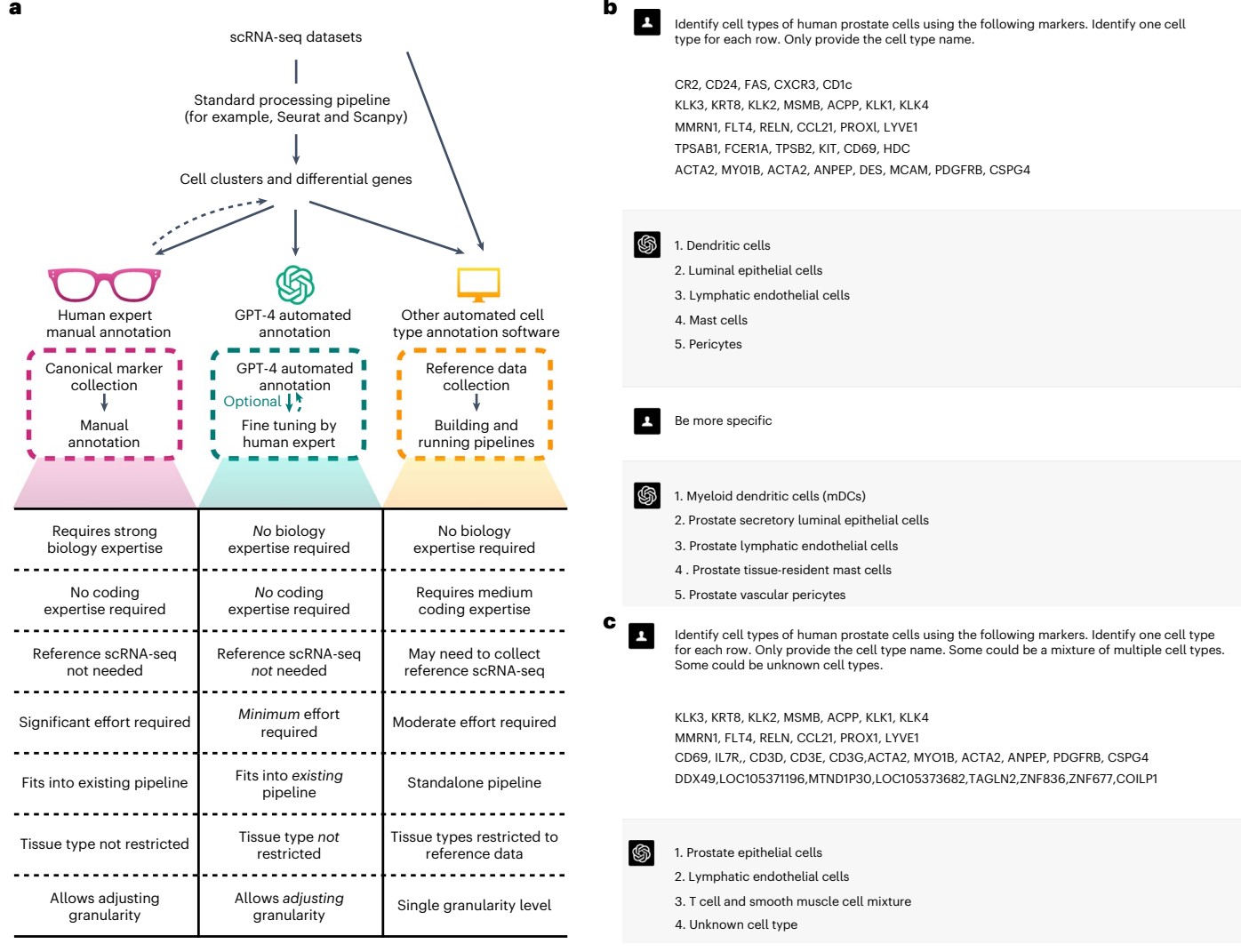

**Fig. 1 | Examples of GPT-4's cell type annotation and comparisons with other methods. a**, Comparison of cell type annotations by human experts, GPT-4, and other automated methods. **b**, Example of GPT-4 annotating human prostate cells with increasing granularity. **c**, Example of GPT-4 annotating single, mixed and new cell types.

annotations more frequently in major cell types (for example, T cells) than in subtypes (for example, CD4 memory T cells), while over 75% of subtypes still achieve full or partial matches (Fig. 2b).

The low agreement between GPT-4 and manual annotations in some cell types does not necessarily imply that GPT-4's annotation is incorrect. For instance, cell types classified as stromal cells include fibroblasts and osteoblasts expressing type I collagen genes, and chondrocytes expressing type II collagen genes. For cells manually annotated as stromal cells, GPT-4 assigns cell type annotations with higher granularity (for example, fibroblasts and osteoblasts), resulting in partial matches and a lower agreement. For cell types that are manually annotated as stromal cells but identified by GPT-4 as fibroblasts or osteoblasts, type I collagen genes show substantially higher expression than type II collagen genes (Fig. 2c). This agrees with the pattern observed in cells manually annotated as chondrocytes, fibroblasts, and osteoblasts (Fig. 2c), suggesting that GPT-4 provides more accurate cell type annotations for stromal cells.

GPT-4 substantially outperforms other methods based on average agreement scores (Methods and Fig. 2d). Using GPTCelltype as the interface, GPT-4 is also notably faster (Fig. 2e), partly due to its utilization of differential genes from the standard single-cell analysis pipelines such as Seurat[3]. Given the integral role of these pipelines, we regard the differential genes as immediately available for GPT-4.

In contrast, other methods like SingleR and ScType require additional steps to reprocess the gene expression matrices. Compared to other methods that are free of charge, GPT-4 incurs a $20 monthly fee for using online web portal. Cost of GPT-4 API is linearly correlated with the number of queried cell types and does not exceed $0.1 for all queries in this study (Fig. 2f).

We further assessed GPT-4's robustness in complex real data scenarios (Fig. 1c) with simulated datasets (Methods). GPT-4 can distinguish between pure and mixed cell types with 93% accuracy, and differentiate between known and unknown cell types with 99% accuracy (Fig. 2g). When the input gene set includes fewer genes or is contaminated with noise, GPT-4's performance decreases but remains high (Fig. 2g). These results demonstrate GPT-4's robustness in various scenarios.

Finally, we assessed the reproducibility of GPT-4's annotations using prior simulation studies (Methods). GPT-4 generated identical annotations for the same marker genes in 85% of cases (Fig. 2h), indicating high reproducibility. Annotations of two GPT-4 versions showed identical agreement scores in most cases, with a Cohen's $\kappa$ of 0.65, demonstrating substantial consistency (Fig. 2i).

While GPT-4 excels in cell type annotation, which surpasses existing methods, there are limitations to consider. Firstly, the undisclosed nature of GPT-4's training corpus makes verifying the basis of

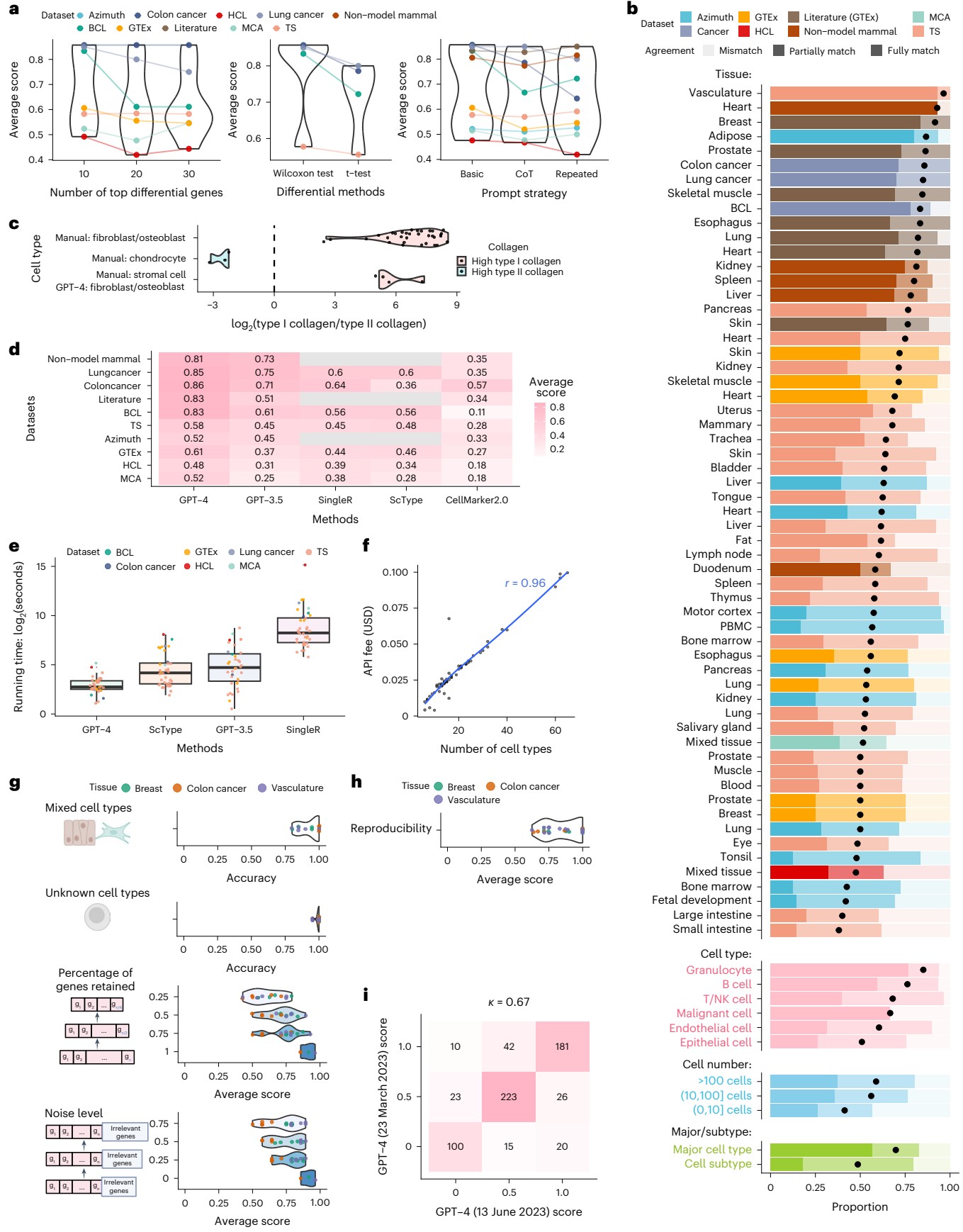

**Fig. 2 | Performance evaluation. a**, Average agreement scores for varying numbers of top differential genes, statistical tests for differential analysis, and prompt strategies. **b**, Proportion of cell types with varying agreement levels in each study and tissue, most abundant broad cell types, malignant cells, different cell population sizes, and major cell types versus cell subtypes. **c**, $\log_2$-transformed ratio of type I (*COL1A1* and *COL1A2*) and II (*COL2A1*) collagen gene expression. **d,e**, Comparison of average agreement scores (**d**) and running times (**e**). In **e**, $n = 59$ for GPT-4 and GPT-3.5 and $n = 36$ for ScType and SingleR. Each boxplot shows the distribution (center: median; bounds of box: first and third quartiles; bounds of whiskers: data points within 1.5× interquartile range from the box; minima; maxima) of running time. **f**, Financial cost of querying GPT-4 API versus cell type numbers. **g**, GPT-4's performance in identifying mixed/single cell types and known/unknown cell types, and under different subsampling and noise levels in multiple simulation rounds (dots). **h**, Reproducibility of GPT-4 annotations. **i**, Consistency of agreement scores between two versions of GPT-4.

---

its annotations challenging, thus requiring human evaluation to ensure annotation quality and reliability. Secondly, human involvement in the optional fine-tuning of the model may affect reproducibility due to subjectivity and could limit the scalability of the model in large datasets. Thirdly, high noise levels in scRNA-seq data and unreliable differential genes can adversely affect GPT-4's annotations. Lastly, over-reliance on GPT-4 risks artificial intelligence hallucination. We recommend validation of GPT-4's cell type annotations by human experts before proceeding with downstream analyses.

While this study focuses on the standard version of GPT-4, fine-tuning GPT-4 with high-quality reference marker gene lists could further improve cell type annotation performance, utilizing services such 'GPTs' provided by OpenAI.

## Online content

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

## Methods

### Dataset collection

For the HuBMAP Azimuth project, manually annotated cell types and their marker genes were downloaded from the Azimuth website (https://azimuth.hubmapconsortium.org/). Azimuth provides cell type annotations for each tissue at different granularity levels. We selected the level of granularity with the fewest number of cell types, provided that there are more than ten cell types within that level. Details of how marker genes were generated are not reported by Azimuth.

For the GTEx[5] dataset, manually annotated cell types, differential gene lists and gene expression matrices were downloaded directly from the publication[5]. In the original study, gene expression raw counts were library-size-normalized and log-transformed after adding a pseudocount of 1 with SCANPY[17]. ComBat[18] was used to account for the protocol- and sex-specific effects with SCANPY[17]. Welch's $t$-test was then performed to identify differential genes that compare one cell type against the rest. For each cell type, genes were ranked increasingly by $P$ values, and genes with the same $P$ values were further ranked decreasingly by $t$-statistics. Top 10, 20 and 30 differential genes were used in this study. Lists of marker genes through literature search and the corresponding cell types were downloaded from the same study[5], and only cell types with at least five marker genes were used.

For the HCL[6] dataset, manually annotated cell types, differential gene lists and the gene expression matrix were downloaded directly from the publication[6]. In the original study, gene expression raw counts underwent a batch removal process to facilitate cross-tissue comparison and were subsequently normalized by library size and log-transformed after adding a pseudocount of 1. Two-sided Wilcoxon rank-sum test was then performed to identify differential genes comparing one cell type against the rest using Seurat[3]. Differential genes were further selected by log fold change larger than 0.25, Bonferroni-adjusted $P$ value smaller than 0.1, and expressed in at least 15% of cells in either population. For each cell type, genes were ranked increasingly by $P$ values, and genes with the same $P$ values were further ranked decreasingly by two-sided Wilcoxon test statistics. Top 10, 20 and 30 differential genes were used in this study.

For the Mouse Cell Atlas (MCA)[7] dataset, manually annotated cell types, differential gene lists and gene expression matrix were downloaded directly from the publication[6]. In the original study, gene expression raw counts underwent a batch removal process to facilitate cross-tissue comparison, and Seurat[3] was used to perform preprocessing and differential analysis. For each cell type, genes were ranked increasingly by $P$ values, and genes with the same $P$ values were further ranked decreasingly by log fold change. Top 10, 20 and 30 differential genes were used in this study.

For non-model mammal dataset[12], manually annotated cell types and lists of marker genes through literature search were downloaded directly from the original study.

For Tabula Sapiens (TS)[8], B-cell lymphoma (BCL)[9], lung cancer[11] and colon cancer[10] datasets, manually annotated cell types and raw gene expression count matrices were downloaded directly from original studies. Raw counts were normalized by library size and log-transformed after adding a pseudocount of 1. Seurat FindAllMarkers() function with default settings was used to obtain differential genes by comparing one cell type with the rest within each tissue. Briefly, genes with at least 0.25 log fold change between two cell populations and detected in at least 10% of cells in either cell population were retained. Two-sided Wilcoxon rank-sum test was then performed for differential analysis. In addition, two-sided two-sample $t$-test was also performed for differential analysis using the FindAllMarkers() function with default settings. For each cell type, genes were ranked increasingly by $P$ values, and genes with the same $P$ values were further ranked decreasingly by log fold changes. Top 10, 20 and 30 differential genes were used in this study.

### Cell type annotation methods

**GPT-4 and GPT-3.5.** All GPT-4 (13 June 2023 version) and GPT-3.5 (13 June 2023 version) cell type annotations in this study were performed using GPTCelltype, an R software package we developed as an interface for GPT models. GPTCelltype takes marker genes or top differential genes as input, and automatically generates prompt message using the following template with the basic prompt strategy:

'Identify cell types of TissueName cells using the following markers separately for each row. Only provide the cell type name. Do not show numbers before the name. Some can be a mixture of multiple cell types.\n GeneList'.

Here 'TissueName' is a variable that will be replaced with the actual name of the tissue (for example, human prostate), and 'GeneList' is a list of marker genes or top differential genes. Genes for the same cell population are joined by comma (,), and gene lists for different cell populations are separated by the newline character (\n). GPT-4 or GPT-3.5 was then queried using the generated prompt message through OpenAI API, and the returned information was parsed and converted to cell type annotations.

For chain-of-thought prompt strategy, the following sentence was added to the beginning of the message generated by the basic prompt strategy: 'Because *CD3* gene is a marker gene of T cells, if *CD3* gene is included in the marker gene list of an unknown cell type, the cell type is likely to be T cells, a subtype of T cells, or a mixed cell type containing T cells'.

For repeated prompt strategy, GPT-4 was queried with the basic prompt strategy repeatedly for five times. The annotation result that appears most frequently among the five queries was selected as the final cell type annotation.

GPT-4 (23 March 2023 version) cell type annotations were performed by manually copying and pasting prompt messages to GPT-4 online web interface (https://chat.openai.com/). The prompt message was constructed using the following template:

'Identify cell types of TissueName cells using the following markers. Identify one cell type for each row. Only provide the cell type name. \n GeneList'.

Computationally identified differential genes in eight scRNA-seq datasets and canonical marker genes identified through literature search in two datasets were used as inputs to GPT-4 and GPT-3.5 (Supplementary Table 2). Cell type annotation for HCL and MCA was performed and evaluated once by aggregating all tissues, similar to the original studies. In other studies, cell type annotation was performed and evaluated within each tissue.

**SingleR.** SingleR[14] (version 1.4.1) R package was used to perform cell type annotations with default settings. For HCL and MCA datasets, the gene expression matrices after batch effect removal, library size normalization and log transformation across all tissues were used as input. For all other datasets, SingleR was performed separately within each tissue, and the input is the log-transformed and library-size normalized gene expression matrix. The built-in Human Primary Cell Atlas reference[19] was used as the reference dataset for all SingleR annotations. SingleR generates single-cell level cell type annotations by returning an assignment score matrix for each single cell and each cell type label in the reference. To convert single-cell level annotations to cell-cluster level annotations, for each manually annotated cell type, we assigned the reference label with assignment scores summed across all single cells in that manually annotated cell type as the predicted cell type annotation.

**ScType.** ScType[15] (version 1.0) R package was used to perform cell type annotations with default settings. To meet the need for computational efficiency when working with large datasets, we developed an in-house version of ScType. We utilized vectorization to optimize the most time-consuming steps, while still generating the same output of the

original ScType software. The input gene expression matrices to ScType were the same as used in SingleR described above. The built-in cell type marker database was used as the reference for all ScType annotations. Manually annotated cell types were treated as cell clusters and given as inputs to ScType. ScType directly generates cluster-level cell type annotations.

**CellMarker2.0.** CellMarker2.0 (ref. 13) only provides an online user interface and does not have a software implementation. We used the exact same marker gene sets or top ten differential gene sets identified by two-sided Wilcoxon tests for GPT-4 and GPT-3.5 cell type annotations as inputs of CellMarker2.0.

### Evaluations of cell type annotations

Cell type annotations by GPT-4 or competing methods were compared to manual annotations provided by the original studies. Each manually or automatically identified cell type annotation was assigned an unambiguous cell ontology (CL) name[20] and a broad cell type name when applicable. A pair of manually and automatically identified cell type annotations was classified as 'fully match' if they have the same annotation term or available CL cell ontology name, 'partially match' if they have the same or subordinate (for example, fibroblast and stromal cell) broad cell type name but different annotations and CL cell ontology names, and 'mismatch' if they have different broad cell type names, annotations and CL cell ontology names.

To facilitate comparison, we assigned agreement scores of 1, 0.5 and 0 to cases of 'fully match', 'partially match' and 'mismatch' respectively, and calculated average scores within each dataset across cell types and tissues.

### Simulation studies and reproducibility

To generate simulation datasets, we used canonical cell type markers through GTEx literature search of human breast cells, the top ten differential genes from the human colon cancer dataset, and the top ten differential genes from the vasculature tissue of the TS dataset as templates. Simulation studies were performed separately for the three tissue types.

To generate simulation datasets of mixed cell types, marker genes for each mixed cell type were created by combining the marker gene lists of two randomly selected cell types. Ten mixed cell types were generated in each simulation iteration. Additionally, we incorporated the original cell type markers of ten randomly chosen cell types as negative controls of single cell types. This entire simulation process was repeated five times. Subsequently, GPT-4 was queried using these simulated marker gene lists, and its performance in differentiating between mixed and single cell types was assessed.

To generate simulation datasets of unknown cell types, we compiled a list of all human genes using the Bioconductor org.Hs.eg.db package[21]. In each simulation iteration, ten simulated unknown cell types were generated. The marker genes for each unknown cell type were produced by combining ten randomly selected human genes. Additionally, we included ten real cell types and their marker genes as negative controls of known cell types, similar to the previous simulation study. This entire simulation process was repeated five times. Subsequently, GPT-4 was queried using these simulated marker gene lists, and its performance in distinguishing between known and unknown cell types was assessed.

To generate simulation datasets with partial marker gene information, we randomly subsampled 25%, 50% or 75% of the original marker genes. The simulation process was repeated five times. Subsequently, GPT-4 was queried using these subsampled marker gene lists, and the performance was assessed by agreement scores.

To generate simulation datasets with contaminated information, we added randomly selected human genes to the original marker gene list. The numbers of randomly selected genes are 25%, 50% or 75% of the number of original marker genes. The simulation process was repeated five times. Subsequently, GPT-4 was queried using these

subsampled marker gene lists, and the performance was assessed by agreement scores.

We assessed the reproducibility of GPT-4 responses by leveraging the repeated querying of GPT-4 with identical marker gene lists of the same negative control cell types in simulation studies. For each cell type, reproducibility is defined as the proportion of instances in which GPT-4 generates the most prevalent cell type annotation. For instance, in the case of vascular endothelial cells, GPT-4 produces 'endothelial cells' eight times and 'blood vascular endothelial cells' once. Consequently, the most prevalent cell type annotation is 'endothelial cells', and the reproducibility is calculated as $\frac{8}{9} = 0.89$.

### GPT-4 API financial cost

According to information provided by OpenAI, the application programming interface (API) cost for running GPT-4 13 June 2023 version is \$0.03 for every thousand input tokens and \$0.06 for every thousand output tokens. For each query, we obtained $i$ and $o$, which represent the numbers of input tokens and output tokens respectively, through the OpenAI API. The total API financial cost is thus calculated as $(0.00003i + 0.00006o)$.

### Reporting summary

Further information on research design is available in the Nature Portfolio Reporting Summary linked to this article.

## Data availability

The data used in this manuscript are all downloaded from publicly available data sources. Specifically, HubMAP Azimuth data were downloaded from the Azimuth website (https://azimuth.hubmapconsortium.org/). GTEx manually annotated cell types and differential gene lists were downloaded from the supplementary materials of the original study[5]. GTEx gene expression matrix was downloaded from the GTEx website (https://gtexportal.org/home/datasets). Marker genes from literature search were downloaded from the supplementary materials of the original study[5]. HCL manually annotated cell types and differential gene lists were downloaded from the supplementary materials of the original study[6]. HCL gene expression matrix was downloaded from figshare (https://figshare.com/articles/dataset/HCL_DGE_Data/7235471). MCA manually annotated cell types and differential gene lists were downloaded from the supplementary materials of the original study[7]. MCA gene expression matrix was downloaded from figshare (https://figshare.com/s/865e694ad06d5857db4b). BCL gene expression matrix and manually annotated cell types were downloaded from Zenodo (https://zenodo.org/record/7813151). Colon cancer gene expression matrix and manually annotated cell types were downloaded from GEO under accession number GSE132465. Lung cancer gene expression matrix and manually annotated cell types were downloaded from GEO under accession number GSE131907. TS gene expression matrix and manually annotated cell types were downloaded from UCSC Cell Browser (https://cells.ucsc.edu/?ds=tabula-sapiens). Marker genes and cell type annotations for the non-model mammal dataset were downloaded from the supplementary materials of the original study[12]. All relevant information about data is described in Methods. All data generated in this study are included in the supplementary tables.

## Code availability

The GPTCelltype package (v.1.0.0) is provided as an open-source software package with a detailed user manual available in the GitHub repository at https://github.com/Winnie09/GPTCelltype. The software is released in Zenodo under https://doi.org/10.5281/zenodo.8317406 for all versions (ref. 22). All codes to reproduce the presented analyses are publicly available in the GitHub repository at https://github.com/Winnie09/GPTCelltype_Paper and also in Zenodo under https://doi.org/10.5281/zenodo.8317410 (https://zenodo.org/record/8317410) (ref. 23). R version 4.0.2 was used to perform the analyses in the manuscript.

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

## Acknowledgements

Z.J. was supported by the National Institutes of Health under award number U54AG075936 and by the Whitehead Scholars Program at Duke University School of Medicine. W.H. was partially supported by the National Institute Of General Medical Sciences of the National Institutes of Health under award number R35GM150887 and by the General Fund at Columbia University Department of Biostatistics.

The content is solely the responsibility of the authors and does not necessarily represent the official views of the National Institutes of Health.

## Author contributions

W.H. and Z.J. conceived the study, conducted the analysis and wrote the manuscript.

## Competing interests

The authors declare no competing interests.

## Additional information

**Correspondence and requests for materials** should be addressed to Wenpin Hou or Zhicheng Ji.

# Reporting Summary

## Statistics

For all statistical analyses, confirm that the following items are present in the figure legend, table legend, main text, or Methods section.

| n/a | Confirmed | |
|---|---|---|
| ☐ | ☒ | The exact sample size (*n*) for each experimental group/condition, given as a discrete number and unit of measurement |
| ☒ | ☐ | A statement on whether measurements were taken from distinct samples or whether the same sample was measured repeatedly |
| ☐ | ☒ | The statistical test(s) used AND whether they are one- or two-sided<br>*Only common tests should be described solely by name; describe more complex techniques in the Methods section.* |
| ☒ | ☐ | A description of all covariates tested |
| ☐ | ☒ | A description of any assumptions or corrections, such as tests of normality and adjustment for multiple comparisons |
| ☐ | ☒ | A full description of the statistical parameters including central tendency (e.g. means) or other basic estimates (e.g. regression coefficient) AND variation (e.g. standard deviation) or associated estimates of uncertainty (e.g. confidence intervals) |
| ☐ | ☒ | For null hypothesis testing, the test statistic (e.g. $F$, $t$, $r$) with confidence intervals, effect sizes, degrees of freedom and $P$ value noted<br>*Give P values as exact values whenever suitable.* |
| ☒ | ☐ | For Bayesian analysis, information on the choice of priors and Markov chain Monte Carlo settings |
| ☒ | ☐ | For hierarchical and complex designs, identification of the appropriate level for tests and full reporting of outcomes |
| ☐ | ☒ | Estimates of effect sizes (e.g. Cohen's *d*, Pearson's *r*), indicating how they were calculated |

*Our web collection on statistics for biologists contains articles on many of the points above.*

## Software and code

Policy information about availability of computer code

| | |
|---|---|
| Data collection | The datasets included in this study were all downloaded directly from the original studies that generate the datasets. No software was used for data collection. |
| Data analysis | Open-source R packages were used to analyze the data in this study. The GPTCelltype package (v.1.0.0) is provided as an open-source software package with a detailed user manual available in Github repository https://github.com/Winnie09/GPTCelltype. The software is released in Zenodo under the accession code DOI: 10.5281/zenodo.8317406 for all versions (https://doi.org/10.5281/zenodo.8317406). SingleR (version 1.4.1), ScType (version 1.0), and CellMarker2.0 were also used for cell type annotation in this study. SingleR was downloaded as an R software package from Bioconductor (https://bioconductor.org/packages/release/bioc/html/SingleR.html). Source code of ScType was obtained from Github (https://github.com/IanevskiAleksandr/sc-type), and an in-house version was implemented for optimized computational efficiency. CellMarker2.0 was accessed directly from the CellMarker2.0 website (http://bio-bigdata.hrbmu.edu.cn/CellMarker/). All codes to reproduce the presented analyses, as well as the in-house version of ScType, are publicly available in Github repository https://github.com/Winnie09/GPTCelltype_Paper and also in Zenodo under the accession code DOI: 10.5281/zenodo.8317410 (https://zenodo.org/record/8317410). R version 4.0.2 was used to perform the analyses in the manuscript. |

For manuscripts utilizing custom algorithms or software that are central to the research but not yet described in published literature, software must be made available to editors and reviewers. We strongly encourage code deposition in a community repository (e.g. GitHub). See the Nature Portfolio guidelines for submitting code & software for further information.

## Data

Policy information about <u>availability of data</u>

All manuscripts must include a <u>data availability statement</u>. This statement should provide the following information, where applicable:

- Accession codes, unique identifiers, or web links for publicly available datasets
- A description of any restrictions on data availability
- For clinical datasets or third party data, please ensure that the statement adheres to our <u>policy</u>

The data used in this manuscript are all downloaded from publicly available data sources. Specifically, HubMAP Azimuth data were downloaded from the Azimuth website (https://azimuth.hubmapconsortium.org/). GTEx manually annotated cell types and differential gene lists were downloaded from the supplementary materials of the original study. GTEx gene expression matrix was downloaded from the GTEx website (https://gtexportal.org/home/datasets). Marker genes from literature search were downloaded from the supplementary materials of the original study. HCL manually annotated cell types and differential gene lists were downloaded from the supplementary materials of the original study. HCL gene expression matrix was downloaded from figshare (https://figshare.com/articles/dataset/HCL_DGE_Data/7235471). MCA manually annotated cell types and differential gene lists were downloaded from the supplementary materials of the original study. MCA gene expression matrix was downloaded from figshare (https://figshare.com/s/865e694ad06d5857db4b). BCL gene expression matrix and manually annotated cell types were downloaded from Zenodo (https://zenodo.org/record/7813151). Colon cancer gene expression matrix and manually annotated cell types were downloaded from GEO under accession number GSE132465. Lung cancer gene expression matrix and manually annotated cell types were downloaded from GEO under accession number GSE131907. TS gene expression matrix and manually annotated cell types were downloaded from UCSC Cell Browser(https://cells.ucsc.edu/?ds=tabula-sapiens). Marker genes and cell type annotations for the non-model mammal dataset were downloaded from the supplementary materials of the original study. All relevant information about data is described in the Methods section. All data generated in this study are included in the supplementary table

## Human research participants

Policy information about <u>studies involving human research participants and Sex and Gender in Research.</u>

| Reporting on sex and gender | NA |
|---|---|
| Population characteristics | NA |
| Recruitment | NA |
| Ethics oversight | NA |

Note that full information on the approval of the study protocol must also be provided in the manuscript.

# Field-specific reporting

Please select the one below that is the best fit for your research. If you are not sure, read the appropriate sections before making your selection.

☒ Life sciences          ☐ Behavioural & social sciences          ☐ Ecological, evolutionary & environmental sciences

For a reference copy of the document with all sections, see nature.com/documents/nr-reporting-summary-flat.pdf

# Life sciences study design

All studies must disclose on these points even when the disclosure is negative.

| Sample size | A total of 10 datasets were included in the study. The Azimuth study contains 11 tissues and 276 cell types. The Human Cell Atlas (HCA) study contains 7 tissues and 72 cell types. The Human Cell Landscape (HCL) study contains 60 tissues and 101 cell types. The literature search dataset contains 7 tissues and 30 cell types. The Mouse Cell Atlas (MCA) study contains 51 tissues and 65 cell types. The lung cancer and colon cancer tissue contain 10 and 7 cell types, respectively. Non-model mammal contains 5 tissue and 24 cell types. GTEx dataset contains 7 tissue and 41 cell types. Tabula sapiens dataset contains 24 tissue and 171 cell types. B cell lymphoma (BCL) dataset contains one tissue and 9 cell types. The number of datasets was determined by the available datasets we can identify that represent a wide variety of tissues. The number of dataset is already sufficiently large since these datasets already represent a wide variety of tissues, cell types, and species so that the conclusions in this study is reliable enough. |
|---|---|
| Data exclusions | For the literature search dataset, only cell types with at least 5 marker genes were used. For other four datasets, all cell types included in the original studies were tested and evaluated in this study. |
| Replication | All simulation studies were repeated five times in this study. |

| | |
|---|---|
| Randomization | This study depends on publicly available data that were already collected and published by other research groups. Randomization is not applicable since no new web-lab experiment is conducted in this study. |
| Blinding | Blinding is not applicable since there is no clinical research data involved in this study. |

# Reporting for specific materials, systems and methods

We require information from authors about some types of materials, experimental systems and methods used in many studies. Here, indicate whether each material, system or method listed is relevant to your study. If you are not sure if a list item applies to your research, read the appropriate section before selecting a response.

## Materials & experimental systems

| n/a | Involved in the study |
|---|---|
| ☒ | ☐ Antibodies |
| ☒ | ☐ Eukaryotic cell lines |
| ☒ | ☐ Palaeontology and archaeology |
| ☒ | ☐ Animals and other organisms |
| ☒ | ☐ Clinical data |
| ☒ | ☐ Dual use research of concern |

## Methods

| n/a | Involved in the study |
|---|---|
| ☒ | ☐ ChIP-seq |
| ☒ | ☐ Flow cytometry |
| ☒ | ☐ MRI-based neuroimaging |

