## [Peer Review File · Nature Methods]

Peer Review Information

Manuscript Title: Assessing GPT-4 for cell type annotation in single-cell RNA-seq analysis

Corresponding author name(s): Zhicheng Ji

Editorial Notes:

Reviewer Comments & Decisions:

Decision Letter, initial version:
--

4th Jun 2023

Dear Dr Ji,

Your Brief Communication, "Reference-free and cost-effective automated cell type annotation with GPT-4 in single-cell RNA-seq analysis", has now been seen by 2 reviewers. As you will see from their comments below, although the reviewers find your work of potential interest, they have raised a number of concerns. We are interested in the possibility of publishing your paper in Nature Methods, but would like to consider your response to these concerns before we reach a final decision on publication.

We therefore invite you to revise your manuscript to address these concerns. After discussion within the editorial team, we suggest you reframe the paper as an assessment of GPT-4 for this application, rather than describing it as a new method. Among other revisions, please substantially improve the validation/benchmarking and provide the software, as Reviewer 1 requests, and sufficiently discuss the limitations that Reviewer 2 points out.

[REDACTED]

We hope to receive your revised paper within 3 months. If you cannot send it within this time, please let us know. In this event, we will still be happy to reconsider your paper at a later date so long as nothing similar has been accepted for publication at Nature Methods or published elsewhere.

OPEN SCIENCE REQUIREMENTS

REPORTING SUMMARY AND EDITORIAL POLICY CHECKLISTS

Please note that these forms are dynamic ‘smart pdfs’ and must therefore be downloaded and completed in Adobe Reader. We will then flatten them for ease of use by the reviewers. If you would like to reference the guidance text as you complete the template, please access these flattened versions at <http://www.nature.com/authors/policies/availability.html>.

DATA AVAILABILITY

All novel DNA and RNA sequencing data, protein sequences, genetic polymorphisms, linked genotype and phenotype data, gene expression data, macromolecular structures, and proteomics data must be deposited in a publicly accessible database, and accession codes and associated hyperlinks must be provided in the “Data Availability” section.

Please include a “Data availability” subsection in the Online Methods. This section should inform readers about the availability of the data used to support the conclusions of your study, including accession codes to public repositories, references to source data that may be published alongside the paper, unique identifiers such as URLs to data repository entries, or data set DOIs, and any other statement about data availability. At a minimum, you should include the following statement: “The data that support the findings of this study are available from the corresponding author upon request”, describing which data is available upon request and mentioning any restrictions on availability. If DOIs are provided, please include these in the Reference list (authors, title, publisher (repository name), identifier, year). For more guidance on how to write this section please see: <http://www.nature.com/authors/policies/data/data-availability-statements-data-citations.pdf>

CODE AVAILABILITY

Please include a “Code Availability” subsection in the Online Methods which details how your custom code is made available. Only in rare cases (where code is not central to the main conclusions of the paper) is the statement “available upon request” allowed (and reasons should be specified).

MATERIALS AVAILABILITY

ORCID

Nature Methods is committed to improving transparency in authorship. As part of our efforts in this direction, we are now requesting that all authors identified as 'corresponding author' on published papers create and link their Open Researcher and Contributor Identifier (ORCID) with their account on the Manuscript Tracking System (MTS), prior to acceptance. This applies to primary research papers only. ORCID helps the scientific community achieve unambiguous attribution of all scholarly contributions. You can create and link your ORCID from the home page of the MTS by clicking on 'Modify my Springer Nature account'. For more information please visit <http://www.springernature.com/orcid>.

Sincerely,

Lin Tang, PhD
Senior Editor
Nature Methods

Reviewers' Comments:

Reviewer #1:

Remarks to the Author:

The authors present an interesting approach to leverage the language model GPT-4 for cell-type annotation in single-cell RNA sequencing (scRNA-seq) studies. This innovative approach aims to address the challenges of cell type identification, which is a critical yet intricate step in scRNA-seq analysis. Although the paper proposes a potentially promising framework for the application of large language models in scRNA-seq analysis, the manuscript currently exhibits significant shortcomings that impede its scientific rigor and innovations. Consequently, at this stage, the paper is not ready for publication.

1. For automatic cell type annotations, a lot of computational tools have been developed and used in many projects. Although the accuracy and efficiency for automatic reference-based annotation still has room to be improved, the critical challenge remains in annotating cell subtypes, rare populations, and cells in disease samples. For example, B lymphoma subsets can hardly be well identified and separated as the signature genes are not well-defined. Instead of showing cases of normal and healthy samples, the authors should also show the ability of GPT-4 in recognizing cell types from diseased samples.
2. The paper's title suggests the GPT-4 approach as being fully automated for cell-type annotation. and the content of the paper, which describes a "semi-automated" process that potentially benefits from human expert input for fine-tuning GPT-4-generated annotations. Although the application of AI can indeed automate much of the process, the need for human intervention in crafting and altering prompts for different situations adds a level of subjectivity to the process. The authors should clarify and discuss the extent of human involvement in the method and discuss how this might impact the repeatability and scalability of their approach.
3. The authors assert that the GPT-4 approach is cost-effective; however, no details or quantitative analysis supporting this claim are provided.
4. The methodology for prompt engineering in this paper lacks robustness. The authors have manually tailored the queries posed to GPT-4, which could potentially introduce bias and inconsistency in the results. Different prompts are used for cell clusters that may be composed of multiple cell types or unknown cell types. While such flexibility may be necessary due to the complexity of biological systems, it simultaneously leaves room for subjective interpretations and decisions.
5. The paper does not provide a software implementation for GPT-4-based cell type annotation. OpenAI provides API access to these models, and a more programmatic and reproducible version of the analysis approach should be made available to the readers.
6. There is a notable absence of a comprehensive analysis of the results. The authors did not offer a comparison between the proposed GPT-4 annotation approach and the existing methods for cell type identification, nor do they compare their method with the GPT-3.5 model, which is freely available.
7. The term 'fully match' implies a perfect alignment between two sets of annotations. However, a detailed examination of the results suggests that this may not always be the case. In several instances, the 'fully match' category includes cases where GPT-4's annotations are similar to, but not identical with, the reference annotations. This discrepancy could lead to an overestimation of the agreement between GPT-4's annotations and the reference, thereby overstating the performance of GPT-4. For example, the supplemental table lists 'fully matched' instances that show discrepancies, such as:
 - ☑ Row 22: CD56-dim Natural Killer - Natural Killer (NK) cells
 - ☑ Row 68: Adrenocortical cells - Steroidogenic cells (e.g., in adrenal cortex or gonads)
8. Although the authors state that the differential gene table is generated by Seurat, the specific parameters or statistical tests used in this analysis are not detailed. Differential gene expression analysis can depend on various factors including normalization method, statistical model, and multiple testing correction approach. Additionally, the rationale for selecting the top 10 differential genes for each cell cluster is not clear, such as whether this number was chosen based on a specific threshold, such as p-value or fold change, or whether it was an arbitrary decision.
9. The author's source code link ('<https://github.com/zji90/gptcelltype>') returns a 404 error.

Reviewer #2:

Remarks to the Author:

In this paper, the authors present an intriguing application of the advanced language model, GPT-4, for cell type annotation in single-cell RNA-seq analysis. They explore the use of GPT-4 to leverage marker gene information and generate accurate cell type annotations, which could potentially streamline the traditionally labor-intensive manual process.

Despite the innovative experiment, the suggested method has severe shortcomings that limit its usefulness in real-world applications:

- 1) The primary concern is that the annotations proposed by GPT-4 cannot be explicitly verified due to the undisclosed specifics of the training corpus utilized to train this model. This lack of transparency is a considerable limitation, making it difficult to critically evaluate the quality and reliability of the generated annotations without manual intervention.
- 2) Even though users can potentially request GPT-4 to provide supporting data for its annotations, the potential for it to invent "hallucinated" references introduces an element of unreliability into the process. The risk of users being misled by fabricated references and the ensuing inaccuracies in the annotation process underscore the limitations of relying predominantly on AI models for scientific research. Without rigorous human validation and interpretation, the reliability of such annotations could be significantly compromised.
- 3) While the proposed method suggests reducing manual effort in cell type annotation, it is crucial to note that users would still need to manually curate the annotations, thereby limiting the practical utility of the study (referred by the authors as "fine tuning by manual experts"). Again, the reliance on human intervention undermines the primary advantage of this approach, which is the automation of the annotation process.
- 4) The presented method's effectiveness heavily relies on the quality of the input - the list of top differential genes. If the input data is incomplete or biased, it might lead to inaccurate or unreliable annotations.
- 5) Another noteworthy and related limitation arises from the observation that the agreement between GPT-4 and manual annotations decreases in marker genes identified by differential analysis. "The agreement decreases in marker genes identified by differential analysis, which may be attributable to a lower proportion of canonical marker genes being identified as top differential genes." Given that differential analysis is a standard approach for finding marker genes in popular single-cell RNA-seq analysis pipelines like Seurat and Scanpy, this limitation significantly impedes the integration of the proposed method with these widely-used pipelines.
- 6) Despite the authors benchmarking the method against 5 datasets, a potential problem lies in the fact that GPT-4 might have used the same training data, which could make the comparison somewhat circular. Without clarity on the specific datasets used in GPT-4's training, we can't rule out the possibility of an overlap, which could overstate the method's effectiveness.
- 7) Given the nature of language models like GPT-4, the annotations generated may vary over different iterations or model versions, introducing inconsistencies in the results.
- 8) The GitHub page linked in the manuscript is not accessible and I was not able to reproduce nor test the validity of the results presented in this manuscript.

In conclusion, while the authors have introduced an intriguing approach, the practical utility of such a method is considerably limited due to the need for extensive manual intervention, the opacity of the model's training, and the potential inconsistencies in the AI-generated results.

Author Rebuttal to Initial comments

Response to Reviewers

We thank the editors and reviewers for their tremendous efforts in reviewing the manuscript. The reviewers have raised valuable comments that are crucial to improving the quality of the manuscript. We have carefully revised the manuscript based on the comments of the reviewers. The revised parts have been marked as blue in the main manuscript. We would like to first provide a summary of major changes in this revised version of manuscript:

- We have added four new datasets for evaluation purposes, including one large dataset (Tabula Sapiens) published after September 2021, and three datasets from cancer samples (BCL, lung cancer, and colon cancer).
- We have included additional competing methods for evaluation purposes, including GPT-3.5, SingleR, ScType, and CellMarker2.0
- We have improved the rigor of the evaluation procedure by identifying the CL cell ontology for each cell type annotation term.
- We have developed a new open-source R software package, GPTCelltype, that automates and streamlines cell type annotation using GPT-4.
- We have changed the title of the manuscript so that it now focuses on assessing the performance of GPT-4 instead of presenting it as a new method.
- We have added additional analysis to evaluate GPT-4's reproducibility and its performance with noisy or subsampled input data.
- We have added discussions of several GPT-4's limitations.

Below are the point-by-point response to the comments.

Response to Editor

After discussion within the editorial team, we suggest you reframe the paper as an assessment of GPT-4 for this application, rather than describing it as a new method.

We thank the editor for this suggestion. We have changed the title of the manuscript to “Assessing GPT-4 for cell type annotation in single-cell RNA-seq analysis”. We have also emphasized the new focus of the paper in the abstract and various places in the main manuscript.

Among other revisions, please substantially improve the validation/benchmarking and provide the software, as Reviewer 1 requests, and sufficiently discuss the limitations that Reviewer 2 points out.

We thank the editor for this comment. As discussed in detail below in point-to-point response to reviewers, we have substantially improved the validation and benchmarking, provided the software, and discussed several limitations of GPT-4.

Data Availability

We have revised the Data Availability section according to the instructions.

Code Availability

We have created a Code Availability according to the instructions.

Response to Reviewer 1

The authors present an interesting approach to leverage the language model GPT-4 for cell-type annotation in single-cell RNA sequencing (scRNA-seq) studies. This innovative approach aims to address the challenges of cell type identification, which is a critical yet intricate step in scRNA-seq analysis. Although the paper proposes a potentially promising framework for the application of large language models in scRNA-seq analysis, the manuscript currently exhibits significant shortcomings that impede its scientific rigor and innovations. Consequently, at this stage, the paper is not ready for publication.

We appreciate the reviewer's thoughtful review of our manuscript. These constructive comments have significantly helped us improve our manuscript. We have provided point-by-point responses to these comments as follows.

1. For automatic cell type annotations, a lot of computational tools have been developed and used in many projects. Although the accuracy and efficiency for automatic reference-based annotation still has room to be improved, the critical challenge remains in annotating cell subtypes, rare populations, and cells in disease samples. For example, B lymphoma subsets can hardly be well identified and separated as the signature genes are not well-defined. Instead of showing cases of normal and healthy samples, the authors should also show the ability of GPT-4 in recognizing cell types from diseased samples.

We thank the reviewer for the constructive suggestion. We have added substantial analysis to assess how well GPT-4 can handle the three challenges: annotating cell subtypes, rare and small cell populations, and cells in disease samples.

First, to evaluate the ability of GPT-4 to annotate cells in disease samples, we included three new studies that are related to cancer samples, including a colon cancer dataset, a lung cancer dataset, and a B cell lymphoma dataset mentioned by the reviewer. The results show that GPT-4 is able to detect the malignant cell populations in both the lung and colon cancer datasets, but failed to detect malignant cells in the B cell lymphoma dataset, as the reviewer have foreseen. We checked the original publication of the B cell lymphoma study and found that the malignant B cells were identified using copy number variable inferred from single-cell RNA-seq data, instead of being directly inferred using marker genes. Thus, this confirms the reviewer's viewpoint that B lymphoma subsets can hardly be well identified potentially due to

a lack of signature gene sets. We also noticed that the Human Cell Landscape (HCL) dataset we included in the previous version of the manuscript also contains cancer samples, in addition to normal samples and

samples with other non-cancer diseases. However, there is no specific malignant cell population reported in the original HCL study.

We have made the following changes to the revised manuscript. First, we added the three new cancer datasets to all applicable analyses in the study. Second, we added a column in Figure 1d indicating if the study contains normal or cancer samples. Third, we specifically discussed the performance of GPT-4 in identifying malignant cell populations in Figure 2c. We have added the following discussions in the revised manuscript:

“Figure 2c also shows the performance of GPT-4 in the five most abundant broad cell types and malignant cells across all studies. GPT-4 shows higher agreement in immune cell types such as granulocytes compared to other cell types such as epithelial cells. GPT-4 was able to identify malignant cells in colon and lung cancer datasets, but failed in the BCL dataset. A potential reason is that there is not a well-defined signature gene set for B lymphoma cells, making the cell type hard to identify. The identification of malignant cells could benefit from other approaches such as copy number variation, as in the original study.”

Second, to evaluate the ability of GPT-4 for annotation cell subtypes, we classified each of the manually identified cell type annotation as a major cell type or a cell subtype. We then examined the performance of GPT-4 in identifying major cell types or cell subtypes separately. The results, shown in Figure 2c, indicate that GPT-4 is able to identify major cell types with high accuracy. The performance decreases when identifying cell subtypes, but there are still more than 75% cases of “fully match” and “partially match” combined. We have also added the following discussions in the revised manuscript:

“Finally, we classified each manually identified cell type annotation as a major cell type (e.g., T cells) or a cell subtype (e.g., CD4 memory T cells). GPT-4 has significantly higher proportion of “fully match” cases in major cell types, although there are still more than 75% cases of “fully match” and “partially match” combined for cell subtypes (Figure 2c).”

Third, to evaluate the ability of GPT-4 for annotation small and rare cell populations, we examined the performance of GPT-4 in cell types with at most 10 cells, with 10 to 100 cells, and with more than 100 cells. The results, shown in Figure 2c, suggests that the performance of GPT-4 drops slightly for small cell populations with at most 10 cells. We have also added the following discussions in the revised manuscript:

“In smaller cell populations comprising no more than 10 cells, GPT-4 exhibits a modest reduction in performance (Figure 2c), which might be attributed to the limited information available in these rare cell groups.”

2. The paper’s title suggests the GPT-4 approach as being fully automated for cell-type annotation. and the content of the paper, which describes a "semi-automated" process that potentially benefits

from human

expert input for fine-tuning GPT-4-generated annotations. Although the application of AI can indeed automate much of the process, the need for human intervention in crafting and altering prompts for different situations adds a level of subjectivity to the process. The authors should clarify and discuss the extent of human involvement in the method and discuss how this might impact the repeatability and scalability of their approach.

We thank the reviewer for the insightful suggestion. We fully agree with the reviewer that the approach becomes semi-automated when human experts are involved in assisting GPT-4 in the cell type annotation process or fine-tuning the results generated by GPT-4. We have made several changes to the manuscript to clarify this important point.

First, we have changed the title of the paper to “Assessing GPT-4 for cell type annotation in single-cell RNA-seq analysis”. This changes the focus of the manuscript from presenting GPT-4 as a new cell type annotation method to assessing GPT-4’s performance in cell type annotation, which is requested by the editor. The new title does not describe GPT-4 as a fully automated approach anymore.

Second, we have changed the way GPT-4 was used to perform cell type annotation in the revised manuscript. In the previous version of manuscript, we manually constructed the prompt messages using the marker gene information, copied and pasted the prompt messages to GPT-4 online user interface, and manually recorded the cell type annotations generated by GPT-4 from the web interface. Thus, as the reviewer pointed out, the previous version of our approach was indeed semi-automated. In this revision, we have developed a software GPTCelltype, which will be discussed in detail below answering the reviewer’s other comments, that uses an automated pipeline to obtain the cell type annotations using GPT-4. The input of the software is the marker gene list or the differential gene list from the Seurat pipeline, and the output of the software is a list of cell type annotations. Thus, we argue that the newly added software GPTCelltype reaches the same level of automation as other existing software such as SingleR and ScType. Using GPTCelltype as an interface for GPT-4 cell type annotation minimizes the human involvement in the process and helps improve the reproducibility of the study. Reassuringly, we found that the results generated by GPTCelltype and with a newer version of GPT-4 (August 3, 2023 version) is highly consistent with the results generated using the previous semi-automated approach and with an older version of GPT-4 (March 23, 2023 version), as reflected in the newly added Figure 2i.

Third, we fully acknowledge that human involvement in the semi-automated process will impact the repeatability and scalability of the process. We have thus added the following discussions to the main manuscript:

“Second, the involvement of human experts in the optional fine-tuning step may negatively impact the repeatability of the results due to the added subjectivity and may reduce the scalability of the approach

when applied to a large number of datasets.”

3. The authors assert that the GPT-4 approach is cost-effective; however, no details or quantitative analysis supporting this claim are provided.

We thank the reviewer for raising this point. In the revised manuscript, we have strengthened the comparison of cost-effectiveness between GPT-4 and other existing methods from two perspectives.

First, we have added a new Supplementary Table 1 that summarizes the requirement of references, the availability of built-in references, the ability to integrate with existing pipelines such as Seurat and Scanpy, and automation levels for GPTCelltype and other existing cell type annotation methods. The results show that most existing cell type annotation methods require reference gene expression data or reference marker gene lists, do not provide comprehensive built-in references, and cannot be directly integrated into standard scRNA-seq processing pipelines such as Seurat and Scanpy. These limitations increase the time and effort required for deploying the pipelines and performing cell type annotations. In comparison, GPT-4 does not require reference gene expression data or reference marker gene lists, and can be directly integrated into Seurat pipeline through the use of GPTCelltype software.

Second, we have added a new Figure 2f comparing the running time of GPT-4 (using GPTCelltype as interface), GPT-3.5 (with GPTCelltype), and existing methods of SingleR and ScType. The results show that GPT-4 (with GPTCelltype) is more computationally efficient compared to other methods. One reason of GPT-4’s computational efficiency is that it utilizes the differential genes generated by standard single-cell processing pipelines such as Seurat. Since running these pipelines is now an integral and inevitable step in analyzing single-cell RNA-seq datasets regardless of how cell type annotation is done, we treat results from these standard pipelines as directly available and did not count the time of running these standard pipelines towards time running GPT-4 (with GPTCelltype). In comparison, existing methods such as SingleR and ScType need to run their own pipelines other than the standard pipelines. These extra steps also involve reprocessing the gene expression matrices, which can be time-consuming. We have added the below discussions in the main manuscript:

“Using GPTCelltype as an interface and working directly on the identified differential genes, GPT-4 is also significantly faster than other methods (Figure 2f). GPT-4’s efficiency is partly attributed to its use of differential genes from standard single-cell analysis pipelines like Seurat. Given the integral role of these pipelines in analyzing scRNA-seq datasets, we regard the differential genes as immediately available for GPT-4. In comparison, other existing methods such as SingleR and ScType need to take extra steps and reprocess the gene expression matrices, which can be computationally intensive. Note that we used an

in-house implementation of ScType (Methods) with significantly increased computational efficiency, since the original ScType implementation cannot process large datasets in reasonable time. We did not compare the running time of CellMarker2.0 since it requires users to manually input gene sets on its online user interface.”

4. The methodology for prompt engineering in this paper lacks robustness. The authors have manually tailored the queries posed to GPT-4, which could potentially introduce bias and inconsistency in the results. Different prompts are used for cell clusters that may be composed of multiple cell types or unknown cell types. While such flexibility may be necessary due to the complexity of biological systems, it simultaneously leaves room for subjective interpretations and decisions.

We thank the reviewer for this comment. We fully agree with the reviewer that using different prompt strategies for different tasks in this study may introduce bias and inconsistency.

In the revised manuscript, we used the same prompt strategy universally for all tasks in the study. The new prompt template is: “Identify cell types of TissueName cells using the following markers separately for each row. Only provide the cell type name. Do not show numbers before the name. Some can be a mixture of multiple cell types. `\n` GeneList”. Here “TissueName” is a variable that will be replaced with the actual name of the tissue (e.g., human prostate), and “GeneList” is a list of marker genes or top differential genes. Genes for the same cell population are joined by comma (,), and gene lists for different cell populations are separated by the newline character (`\n`). The prompt messages were automatically generated by GPTCelltype using the above template. In the revised manuscript, we have shown that the new prompt strategy works well for various tasks performed in the study.

Instead of manually performing cell type annotation using the online web interface of GPT-4 in the previous version of manuscript, we used GPTCelltype software interface we developed to automatically generate the prompts and then feed the prompts to GPT-4 API in the revised manuscript. We believe that using a universal prompt strategy and a software interface for GPT-4 cell type annotation has significantly reduced bias and inconsistency in the study.

We have added below descriptions in the main manuscript:

“All GPT-4 (August 3, 2023 version) and GPT-3.5 (August 3, 2023 version) cell type annotations in this study were performed using GPTCelltype, an R software package we developed as an interface for GPT models. GPTCelltype takes marker genes or top differential genes as input, and automatically generates prompt message using the following template:

“Identify cell types of TissueName cells using the following markers separately for each row. Only

provide the cell type name. Do not show numbers before the name. Some can be a mixture of multiple

cell types. `\n GeneList`".

Here "TissueName" is a variable that will be replaced with the actual name of the tissue (e.g., human prostate), and "GeneList" is a list of marker genes or top differential genes. Genes for the same cell population are joined by comma (,), and gene lists for different cell populations are separated by the newline character (`\n`). GPT-4 or GPT-3.5 was then queried using the generated prompt message through OpenAI API, and the returned information was parsed and converted to cell type annotations."

5. The paper does not provide a software implementation for GPT-4-based cell type annotation. OpenAI provides API access to these models, and a more programmatic and reproducible version of the analysis approach should be made available to the readers.

We thank the reviewer for this very helpful suggestion. We have developed a publicly available R software package, "GPTCelltype", for users to conveniently access the GPT-4-based cell type annotation. It can be freely accessed on Github (<https://github.com/Winnie09/GPTCelltype>) with a user-friendly online manual.

GPTCelltype allows two types of inputs. The first type of input is a differential gene table obtained after running the standard Seurat pipeline. The second type of input is a list of marker genes or differential genes prepared by the user to allow for flexibility. The user has the option to choose the number of top differential genes to be used if a Seurat differential gene table is given as the input. The user also has the option to choose the specific GPT model provided by OpenAI (default is GPT-4).

GPTCelltype will then automatically generate the prompt using the prompt strategy discussed above. If the user provides an optional OpenAI key, which is required to access the OpenAI API for GPT models, GPTCelltype will directly feed the generated prompt to the OpenAI API, and convert the returned output into a vector of cell types. GPTCelltype provides additional instructions on how to assign the cell types back to the original Seurat object by creating an extra column of annotated cell types in the metadata slot. This streamlines the cell type annotation process and allows users to directly perform all Seurat downstream analyses, such as visualizing different cell types on a UMAP space. If the OpenAI key is not provided, GPTCelltype will directly output the prompt message so users can copy and paste the prompt message to the GPT-4 or ChatGPT online web interface. This flexibility also allows users to further interact with GPT-4 or ChatGPT to fine-tune the clustering results.

We provided a comprehensive vignette demonstrating the usage of GPTCelltype in various use cases. We have also provided detailed descriptions and screenshots of how to obtain OpenAI keys on the OpenAI website.

6. There is a notable absence of a comprehensive analysis of the results. The authors did not offer a comparison between the proposed GPT-4 annotation approach and the existing methods for cell type identification, nor do they compare their method with the GPT-3.5 model, which is freely available.

We thank the reviewer for raising this very important point. We fully agree with the reviewer that a comprehensive analysis and comparison are needed to fully understand the performance of GPT-4 in cell type annotation. We have made the following changes in the revised manuscript.

First, we included a new Supplementary Table 1 that compares the requirement of reference data, the ability to integrate with existing pipelines such as Seurat and Scanpy, and automation level for GPT-4 (with GPTCelltype) and other existing cell type annotation methods.

Second, we picked GPT-3.5 and three other existing cell type annotation methods (SingleR, ScType, and CellMarker2.0) for an in-depth and comprehensive comparison of performance. We picked the three methods (SingleR, ScType, and CellMarker2.0) because they provide options for cell type annotation in a wide range of tissues, while other methods do not offer reference data for many tissues evaluated in this study (Supplementary Table 1). The agreement scores between manually and automatically generated cell type annotations are shown in the newly added Figure 2e, and the comparison of running time is shown in the newly added Figure 2f. These results suggest that GPT-4 significantly outperforms all other competing methods, including its prior version GPT-3.5, in both agreement scores and computational efficiency. We have also added the below discussions in the main manuscript:

“We next compared the performance of GPT-4 with other competing methods by averaging the agreement scores within each study across all tissues and cell types (Figure 2e). In all datasets, GPT-4 performs significantly better than other methods, including its prior version GPT-3.5. GPT-3.5 has similar performance compared to SingleR, ScType and CellMarker2.0. Using GPTCelltype as an interface and working directly on the identified differential genes, GPT-4 is also significantly faster than other methods (Figure 2f). GPT-4’s efficiency is partly attributed to its use of differential genes from standard single-cell analysis pipelines like Seurat. Given the integral role of these pipelines in analyzing scRNA-seq datasets, we regard the differential genes as immediately available for GPT-4. In comparison, other existing methods such as SingleR and ScType need to take extra steps and reprocess the gene expression matrices, which can be computationally intensive. Note that we used an in-house implementation of ScType (Methods) with significantly increased computational efficiency, since the original ScType implementation cannot process large datasets in reasonable time. We did not compare the running time of CellMarker2.0 since it requires users to manually input gene sets on its online user interface.”

Third, we have added other analysis results in the revised manuscript, including the performance of GPT-4 in malignant cell populations, in cell populations with different numbers of cells, and in major cell types or cell subtypes, which we have discussed above. We have also included the performance of GPT-4 when there is only a subset of marker gene information available (Figure 2g) and when the marker gene information is contaminated (Figure 2g). We have also shown that cell type annotations by different versions of GPT-4 are highly consistent (newly added Figure 2i). We believe these additional results have made the assessment of GPT4 more compelling.

7. The term 'fully match' implies a perfect alignment between two sets of annotations. However, a detailed examination of the results suggests that this may not always be the case. In several instances, the 'fully match' category includes cases where GPT-4's annotations are similar to, but not identical with, the reference annotations. This discrepancy could lead to an overestimation of the agreement between GPT-4's annotations and the reference, thereby overstating the performance of GPT-4. For example, the supplemental table lists 'fully matched' instances that show discrepancies, such as: Row 22: CD56-dim Natural Killer - Natural Killer (NK) cells Row 68: Adrenocortical cells - Steroidogenic cells (e.g., in adrenal cortex or gonads)

We thank the reviewer for this helpful comment. We agree with the reviewer that some “fully match” cases were not well defined in the previous version of manuscript, as the reviewer pointed out. In the revised manuscript, we have substantially improved how the agreement between manually and automatically generated cell type annotation terms are determined, and redone all the evaluations based on the improved scoring method.

Specifically, for each cell type annotation term, including manual annotations obtained from original studies and annotations generated by automatic methods such as GPT-4, we assigned an unambiguous CL cell ontology name by searching EMBL-EBI OLS CL cell ontology website (see the website here: <https://www.ebi.ac.uk/ols/ontologies/cl>). The assignment of CL cell ontology name was only done for applicable terms, and we did not assign CL ontology name if no unambiguous CL cell ontology name is available. For example, annotation terms such as “cycling cells” do not refer to any specific CL cell ontology name. We also assign a broad cell type name for each cell type annotation term. For example, “t/nk cell” is a broad cell type name assigned to the annotation term “CD4 T cells”. We then define “fully match” if a pair of cell type annotations have the same annotation terms, or the same CL cell ontology name if available. We define “partially match” if a pair of cell type annotations have the same or subordinate

(e.g., fibroblast and stromal cell) broad cell type name, but different annotation terms or CL cell ontology

names. We define “mismatch” if none of the annotation terms, CL cell ontology names, or broad cell type names agree for a pair of cell types.

We have added the below descriptions in the main manuscript for the revised agreement scoring method:

“Specifically, each manually or automatically identified cell type annotation was assigned an unambiguous CL cell ontology name and a broad cell type name when applicable. A pair of manually and automatically identified cell type annotations was classified as “fully match” if they have the same annotation term or available CL cell ontology name, “partially match” if they have the same or subordinate (e.g., fibroblast and stromal cell) broad cell type name but different annotations and CL cell ontology names, and “mismatch” if they have different broad cell type names, annotations, and CL cell ontology names.”

8. Although the authors state that the differential gene table is generated by Seurat, the specific parameters or statistical tests used in this analysis are not detailed. Differential gene expression analysis can depend on various factors including normalization method, statistical model, and multiple testing correction approach. Additionally, the rationale for selecting the top 10 differential genes for each cell cluster is not clear, such as whether this number was chosen based on a specific threshold, such as p-value or fold change, or whether it was an arbitrary decision.

We appreciate the reviewer for pointing this out. We fully agree with the reviewer that different parameters and statistical tests may lead to different results in differential gene expression analysis. In the revised manuscript, we have provided details of how differential tests were done and how differential genes were generated and ranked. The following contents are added to the methods section of the revised manuscript:

“For GTEx dataset, manually annotated cell types, differential gene lists, and gene expression matrices were downloaded directly from GTEx. In the original study, gene expression raw counts were library size normalized and log-transformed after adding a pseudocount of 1 with SCANPY. ComBat was used to account for the protocol- and sex-specific effects with SCANPY. Welch’s t-test was then performed to identify differential genes comparing one cell type against the rest. For each cell type, genes were ranked increasingly by p-values, and genes with the same p-values were further ranked decreasingly by t-statistics. Top 10, 20, and 30 differential genes were used in this study. Lists of marker genes through literature search and the corresponding cell types were downloaded from the same study, and only cell types with at least 5 marker genes were used.

For HCL dataset, manually annotated cell types, differential gene lists, and gene expression matrix

were downloaded directly from HCL. In the original study, gene expression raw counts underwent a batch removal process to facilitate cross-tissue comparison, and were subsequently library size normalized and log-transformed after adding a pseudocount of 1. Two-sided Wilcoxon rank-sum test was then performed to identify differential genes comparing one cell type against the rest using Seurat. Differential genes were further selected by log foldchange larger than 0.25, Bonferroni-adjusted p-value smaller than 0.1, and expressed in at least 15% of cells in either population. For each cell type, genes were ranked increasingly by p-values, and genes with the same p-values were further ranked decreasingly by wilcoxon test statistics. Top 10, 20, and 30 differential genes were used in this study.

For MCA dataset, manually annotated cell types, differential gene lists, and gene expression matrix were downloaded directly from MCA. In the original study, gene expression raw counts underwent a batch removal process to facilitate cross-tissue comparison, and Seurat was used to perform preprocessing and differential analysis. For each cell type, genes were ranked increasingly by p-values, and genes with the same p-values were further ranked decreasingly by log fold changes. Top 10, 20, and 30 differential genes were used in this study.

For TS, BCL, lung cancer, and colon cancer datasets, manually annotated cell types and raw gene expression count matrices were downloaded directly from original studies. Raw counts were normalized by library size and log-transformed after adding a pseudocount of 1. Seurat FindAllMarkers() function with default settings was used to obtain differential genes by comparing one cell type with the rest within each tissue. Briefly, genes with at least 0.25 log fold change between two cell populations and detected in at least 10% of cells in either cell populations were retained. Two-sided Wilcoxon rank-sum test was then performed for differential analysis. For each cell type, genes were ranked increasingly by p-values, and genes with the same p-values were further ranked decreasingly by log fold changes. Top 10, 20, and 30 differential genes were used in this study.”

In the revised manuscript, we have provided the rationale for selecting the top 10 differential genes. To assess how different numbers of differential genes may impact the performance of cell type annotation by GPT-4, we used the top 10, 20, and 30 differential genes as inputs to GPT-4, and evaluated the performance based on the average agreement score when comparing to the manual annotations in original studies. The results, as shown in Figure 2b in the revised manuscript, suggest that using the top 10 differential genes leads to the best agreement score, and the agreement score decreases slightly with larger numbers of differential genes used. Thus, we used the top 10 differential genes for all subsequent analyses for GPT-4.

9. The author’s source code link (<https://github.com/zji90/gptcelltype’>) returns a 404 error.

We thank the reviewer for pointing this out. We are very sorry that the source code link provided in the previous version of the manuscript does not work. We have fixed the link and reuploaded all code used in this study to the following GitHub repository:

https://github.com/Winnie09/GPTCelltype_Paper

We have also released a GitHub repository for the software, GPTCelltype, that we developed specifically for this paper to facilitate automated cell type annotations using GPT models. The link is:

<https://github.com/Winnie09/GPTCelltype>

Response to Reviewer 2

In this paper, the authors present an intriguing application of the advanced language model, GPT-4, for cell type annotation in single-cell RNA-seq analysis. They explore the use of GPT-4 to leverage marker gene information and generate accurate cell type annotations, which could potentially streamline the traditionally labor-intensive manual process. Despite the innovative experiment, the suggested method has severe shortcomings that limit its usefulness in real-world applications:

We appreciate the reviewer's thoughtful review of our manuscript. These constructive comments have significantly helped us improve our manuscript. We have provided point-by-point responses to these comments as follows.

1) The primary concern is that the annotations proposed by GPT-4 cannot be explicitly verified due to the undisclosed specifics of the training corpus utilized to train this model. This lack of transparency is a considerable limitation, making it difficult to critically evaluate the quality and reliability of the generated annotations without manual intervention.

We thank the reviewer for raising this important point. We concur with the reviewer regarding the transparency limitations due to the undisclosed specifics of the training corpus. As the reviewer mentioned, the training corpus of GPT-4 is largely undisclosed. Thus, it is not possible to evaluate the validity and reliability of materials GPT-4 used to obtain the knowledge of marker genes and cell type annotations. If the training materials are biased, low-quality, or not comprehensive enough, the quality of the cell type annotation by GPT-4 will also be influenced. Due to this lack of transparency, in this study, we were only able to manually evaluate if the end product of cell type annotations matched with annotations by human experts. We have added the below discussions in the main manuscript to state this limitation:

“Although GPT-4 has a strong performance in cell type annotation and outperforms existing methods according to our assessment, users should still be aware of several limitations when applying GPT-4 for cell type annotation. First, unlike other cell type annotation methods, the training corpus of GPT-4 is largely undisclosed, making it difficult to explicitly verify the basis upon which GPT-4 generates annotations. Certain human effort can still be needed to critically evaluate the quality and reliability of the annotations generated by GPT-4.”

2) Even though users can potentially request GPT-4 to provide supporting data for its annotations, the

potential for it to invent "hallucinated" references introduces an element of unreliability into the process. The risk of users being misled by fabricated references and the ensuing inaccuracies in the annotation process underscore the limitations of relying predominantly on AI models for scientific research. Without rigorous human validation and interpretation, the reliability of such annotations could be significantly compromised.

We thank the reviewer for this constructive suggestion. The evaluation results presented in the revised manuscript show that GPT-4 is able to generate reliable results in most cases and outperforms existing methods in both accuracy and efficiency. Nevertheless, we agree with the reviewer that the possibility of AI hallucinations and GPT models generating misleading results cannot be ruled out, and blindly trusting the results generated by GPT-4 can lead to unforeseen consequences in downstream analyses. We have thus added the following discussion in the main manuscript:

“Finally, predominantly relying on GPT-4 for cell type annotations could be risky in the case of AI hallucination. It is recommended that human experts confirm the validity of cell type annotations generated by GPT-4 before conducting downstream analyses.”

We have also made changes to the GPTCelltype, the software tool we developed for annotating cell types using GPT-4, to warn users of potential risks when predominately relying on the results from GPT-4. The below message has been included in the vignette page of the software:

It is always recommended to check the results returned by GPT-4 in case of AI hallucination, before going to downstream analysis.

3) While the proposed method suggests reducing manual effort in cell type annotation, it is crucial to note that users would still need to manually curate the annotations, thereby limiting the practical utility of the study (referred by the authors as “fine tuning by manual experts”). Again, the reliance on human intervention undermines the primary advantage of this approach, which is the automation of the annotation process.

We thank the reviewer for providing this very useful comment. We agree with the reviewer that in the previous version of the manuscript, considerable human effort is still needed to interact with the GPT-4 online user interface to complete the cell type annotation procedure. In the revised manuscript, we have developed a publicly available R software package, "GPTCelltype", for users to conveniently access the GPT-4-

based cell type annotation. It can be freely accessed on Github (<https://github.com/Winnie09/GPTCelltype>)

with a user-friendly online manual. We believe that GPTCelltype significantly improves the automation and usability level of the approach.

GPTCelltype allows two types of inputs. The first type of input is a differential gene table obtained after running the standard Seurat pipeline. The second type of input is a list of marker genes or differential genes prepared by the user to allow for flexibility. The user has the option to choose the number of top differential genes to be used if a Seurat differential gene table is given as the input. The user also has the option to choose the specific GPT model provided by OpenAI (default is gpt-4).

GPTCelltype will then automatically generate the prompt using the prompt strategy discussed above. If the user provides an optional OpenAI key, which is required to access the OpenAI API for GPT models, GPTCelltype will directly feed the generated prompt to the OpenAI API, and convert the returned output into a vector of cell types. GPTCelltype provides additional instructions on how to assign the cell types back to the original Seurat object by creating an extra column of annotated cell types in the metadata slot. This streamlines the cell type annotation process and allows users to directly perform all Seurat downstream analyses, such as visualizing different cell types on a UMAP space. If the OpenAI key is not provided, GPTCelltype will directly output the prompt message so users can copy and paste the prompt message to the GPT-4 or ChatGPT online web interface. This flexibility also allows users to further interact with GPT-4 or ChatGPT to fine-tune the clustering results.

We provided a comprehensive vignette demonstrating the usage of GPTCelltype in various use cases. We have also provided detailed descriptions and screenshots of how to obtain OpenAI keys on the OpenAI website.

Note that as discussed previously, it is still recommended that human experts confirm the validity of cell type annotation results returned by GPT-4 and GPTCelltype. However, the introduction of GPTCelltype software has significantly reduced the level of human effort compared to the approach in the previous version of the manuscript.

4) The presented method's effectiveness heavily relies on the quality of the input - the list of top differential genes. If the input data is incomplete or biased, it might lead to inaccurate or unreliable annotations.

We thank the reviewer for this helpful suggestion. We agree with the reviewer that in real applications, input gene sets may contain incomplete or biased information. To address this comment, we have included two new simulation studies in the revised manuscript.

In the first simulation study, we randomly subsampled 25%, 50%, and 75% of human breast tissue

marker genes obtained from the “literature search” dataset. These subsampled gene sets represent

incomplete input data. We then evaluated the performance of GPT-4 cell type annotation using the subsampled gene sets. The results, as shown in the new Figure 2g, show that the performance decreases with a smaller number of genes in the input data. However, the level of decrease is small and GPT-4 still maintains a relatively high performance even when there are only half of the genes.

In the second simulation study, we added different numbers of randomly selected human genes (considered as noise) to the human breast tissue marker genes obtained from the literature search dataset, mimicking a situation where the input data is contaminated or biased. The results, as shown in the new Figure 2g, suggest that the performance only decreases slightly with a higher level of contamination in the data.

In general, we conclude that GPT-4's performance is not significantly impacted when the input data is incomplete or biased. We have added the following discussions to the main manuscript as well:

“We then tested the performance of GPT-4 with partial marker gene information by randomly subsampling 75%, 50%, and 25% of original marker genes (Methods). GPT-4's performance decreases with a smaller number of marker genes but still maintains at a high level even with half of the original marker genes available (Figure 2g). Finally, we evaluated the performance of GPT-4 when the input information is contaminated by adding random genes to the marker gene list (Methods). GPT-4's performance only decreases slightly with a higher level of contamination and maintains at a high level (Figure 2g). All these results suggest that GPT-4 has a robust performance when dealing with different situations.”

5) Another noteworthy and related limitation arises from the observation that the agreement between GPT-4 and manual annotations decreases in marker genes identified by differential analysis. “The agreement decreases in marker genes identified by differential analysis, which may be attributable to a lower proportion of canonical marker genes being identified as top differential genes.” Given that differential analysis is a standard approach for finding marker genes in popular single-cell RNA-seq analysis pipelines like Seurat and Scanpy, this limitation significantly impedes the integration of the proposed method with these widely-used pipelines.

We thank the reviewer for raising this point. We agree with the reviewer that the quality of the differential analysis will influence the performance of GPT-4 cell type annotation. However, we would like to point out that the performance decrease will not significantly impact how GPT-4 can be integrated into existing pipelines.

Firstly, we note that while GPT-4's performance with differential genes is reduced compared to its optimal results using canonical marker genes from literature, it remains commendably high. In fact, GPT-4

annotations still fully or partially match manual annotations for at least 75% of cell types in almost all

tissue types when using differential genes as input. Thus, the decreased performance implies a switch from superior performance (70% fully match for canonical marker genes) to a relatively worse, but still satisfactory performance (75% fully or partially match for differential genes). In real practice, GPT-4 can already generate reasonable cell type annotations in most cases and can thus be integrated into existing pipelines. In addition, our newly developed software, GPTCelltype, integrates GPT-4 into the Seurat pipeline to streamline the cell type annotation. To help readers better understand this point, we have rewritten the related sentences in the main manuscript with a different tone:

“Notably, the agreement is especially pronounced for marker genes identified by literature searches, where GPT-4’s annotations fully match with manual annotations in 70% of cases. While there is a decrease in agreement for marker genes identified by differential analysis, the agreement remains commendably high, rendering GPT-4 suitable for a broad range of datasets.”

Second, we would like to argue that the relatively worse performance of GPT-4 when using differential genes as input is largely due to the high level of sparsity and noise in the single-cell RNA-seq data itself. It is well known in computer science and related fields that the quality of the output of a system is largely determined by the quality of the input to the system. As a result, differential analysis, which is often conducted by some statistical or computational methods, will yield suboptimal differential genes that may not fully represent the identity of the cell population if the input scRNA-seq data is noisy, which is often the case in real practice. The suboptimal differential genes will further lead to suboptimal performance of cell type annotation methods such as GPT-4. In comparison, canonical marker genes obtained by literature search are not affected by the high level of noise in scRNA-seq data and thus have better performance. This limitation will be alleviated if scRNA-seq data and the differential genes are less noisy. We have added the below limitations to the main manuscript:

“Third, a high level of noise in scRNA-seq data and unreliable differential genes may negatively impact GPT-4’s cell type annotations.”

6) Despite the authors benchmarking the method against 5 datasets, a potential problem lies in the fact that GPT-4 might have used the same training data, which could make the comparison somewhat circular. Without clarity on the specific datasets used in GPT-4’s training, we can’t rule out the possibility of an overlap, which could overstate the method’s effectiveness.

We thank the reviewer for raising this point. We agree with the reviewer that the comparison could be circular if the benchmark datasets were included as GPT-4’s training data. We would like to address this

comment in the following ways.

First, we would like to clarify that the details of the training corpus by GPT-4 are not fully undisclosed. Although the exact materials GPT-4 used to train its model are largely unknown, it is disclosed by OpenAI that GPT-4 has a knowledge cutoff of September 2021, as stated in GPT-4 technical report published by OpenAI (<https://cdn.openai.com/papers/gpt-4.pdf>). Thus, datasets published after September 2021 are not included as part of GPT-4's training corpus. Evaluating GPT-4's performance in cell type annotation using datasets published after September 2021 will enhance the evaluation's rigor and reveal its adaptability to new gene sets not present in its training corpus.

To this end, we have added two new datasets that were published after September 2021 in the revised manuscript. The first dataset, Tabula Sapiens (TS), is a large dataset that contains 24 tissues and 171 cell types. The second dataset, B cell lymphoma, is a small dataset from cancer samples and includes malignant cells. Together with the two datasets already included in the previous version of manuscript that were also published after September 2021 (GTEx and literature), the revised manuscript now includes four datasets that were published after September 2021, covering a wide range of tissues and cell types. We believe that this has substantially improved the evaluation rigor.

Second, we have added discussions in the main manuscript to raise awareness that the evaluation results for datasets published before September 2021 should be interpreted with care.

“It is worth noting that the Azimuth, HCL, MCA, lung cancer, and colon cancer datasets were published before September 2021 (Figure 1d), the cutoff date of GPT-4's training corpus. Thus, the results of these datasets should be interpreted with care since they may not fully reflect the performance of GPT-4 when dealing with new gene sets not in the training corpus.”

7) Given the nature of language models like GPT-4, the annotations generated may vary over different iterations or model versions, introducing inconsistencies in the results.

We thank the reviewer for raising this critical point. We agree with the reviewer that GPT-4 response may vary over different iterations and model versions. We have performed the following two analyses to address this comment.

First, we incorporated a reproducibility analysis to compare GPT-4's outputs when presented with identical gene sets repeatedly (i.e., the same prompt). Although this analysis appeared in our prior manuscript, it has been updated in this revision to reflect results from the newer GPT-4 version. The findings indicate that GPT-4 produced consistent annotations for the same cell type markers in 88.3% of instances on average, as depicted in Figure 2h.

Second, we compared cell type annotations between two GPT-4 versions: the March 23, 2023 version,

referenced in our previous manuscript, and the August 3, 2023 version, addressed in this revised manuscript. As illustrated in the newly added Figure 2i, both versions predominantly exhibit equivalent agreement scores, achieving a Cohen's Kappa of 0.65, indicating consistent performance across GPT-4 versions in cell type annotation.

In conclusion, the analysis results show that annotations are generally consistent across different iterations and model versions of GPT-4. We have added the below descriptions about reproducibility in the main manuscript:

“We also evaluated the reproducibility of GPT-4 annotations leveraging results in previous simulation studies (Methods). On average, GPT-4 generated identical annotations for the same cell type markers in 88.3% of cases (Figure 2h), showing a high level of reproducibility. In addition, we compared the annotations between an older version of GPT-4 (March 23, 2023 version) and the August 3, 2023 version of GPT-4. The two versions of GPT-4 have the same agreement score in most cases, and reach a Cohen's Kappa of 0.65, showing a substantial consistency (Figure 2i).”

8) The GitHub page linked in the manuscript is not accessible and I was not able to reproduce nor test the validity of the results presented in this manuscript.

We thank the reviewer for pointing this out. We are very sorry that the source code link provided in the previous version of the manuscript does not work. We have fixed the link and reuploaded all code used in this study to the following GitHub repository:

https://github.com/Winnie09/GPTCelltype_Paper

We have also released a GitHub repository for the software, GPTCelltype, that we developed specifically for this paper to facilitate automated cell type annotations using GPT models. The link is:

<https://github.com/Winnie09/GPTCelltype>

In conclusion, while the authors have introduced an intriguing approach, the practical utility of such a method is considerably limited due to the need for extensive manual intervention, the opacity of the model's training, and the potential inconsistencies in the AI-generated results.

We thank the reviewer for providing the overall comments. We believe that in the revised version of the manuscript, the concern of extensive manual intervention has mainly been addressed by the introduction of the software GPTCelltyp, the inconsistencies have been addressed by additional reproducibility analysis,

and we have included extensive discussions of GPT-4's limitations as suggested by the reviewer.

Decision Letter, first revision:

2nd Nov 2023

Dear Dr Ji,

Thank you for your letter detailing how you would respond to the reviewer concerns regarding your Brief Communication, "Assessing GPT-4 for cell type annotation in single-cell RNA-seq analysis". We have decided to invite you to revise your manuscript, before we reach a final decision on publication.

[REDACTED]

We hope to receive your revised paper within 6 weeks. If you cannot send it within this time, please let us know. In this event, we will still be happy to reconsider your paper at a later date so long as nothing similar has been accepted for publication at Nature Methods or published elsewhere.

OPEN SCIENCE REQUIREMENTS

REPORTING SUMMARY AND EDITORIAL POLICY CHECKLISTS

DATA AVAILABILITY

CODE AVAILABILITY

Please include a "Code Availability" subsection in the Online Methods which details how your custom code is made available. Only in rare cases (where code is not central to the main conclusions of the paper) is the statement "available upon request" allowed (and reasons should be specified).

For more information on our code sharing policy and requirements, please see: <https://www.nature.com/nature-research/editorial-policies/reporting-standards#availability-of-computer-code>

MATERIALS AVAILABILITY

ORCID

Sincerely,

Lin Tang, PhD
Senior Editor
Nature Methods

Reviewers' Comments:

Reviewer #1:

Remarks to the Author:

I appreciate the authors' efforts to address some of the concerns raised in the initial review. The revised manuscript has refocused on GPT-4's assessment in cell type annotation tasks from scRNA-seq, improving the methodology, developing the GPTCelltype R package, and providing more analyses. However, I still have the following concerns:

1. The authors have not thoroughly assessed various prompt engineering strategies, which are crucial for utilizing large language models like GPT-4. These strategies can significantly influence the model's performance, and it is currently unclear whether the observed improvements in GPT-4's performance are due to the model itself or the specific prompt engineering strategy employed. A systematic assessment of different prompt engineering strategies, including comparisons of prompt formulations, data preprocessing approaches, and input/output encoding methods, is needed to clarify the factors contributing to the performance improvement. Moreover, the authors should provide a detailed discussion of the prompt engineering strategies tested and the rationale behind their final choice to help readers understand the impact of prompt engineering on GPT-4's performance in cell type annotation and make informed decisions when employing GPTCelltype or similar approaches in their own research. Without a systematic assessment of prompt engineering strategies, it remains unclear whether the reported performance improvements are a direct result of GPT-4's capabilities or an artifact of the specific prompt formulation employed in this study.

2. While the authors' new analysis provides a more comprehensive assessment of GPT-4's performance in various challenging scenarios, it does not directly address the concern of overfitting. Overfitting occurs when a model adapts too well to the training data and fails to perform well on

unseen data. The simulation studies described in the revised manuscript indeed evaluate GPT-4's performance in differentiating mixed and single cell types, distinguishing known and unknown cell types, and handling partial or contaminated marker gene information. However, these analyses do not directly address the concern of overfitting and the generalizability of GPT-4's performance to new, unseen data. The analysis on known and unknown cell types does not provide a direct measure of GPT-4's performance on new data that could include cell types from different species, tissues, or experimental conditions not seen during the training phase.

3. The authors have made efforts to evaluate the influence of input data size and quality on GPT-4's performance by conducting simulation studies on human breast cancer data. Despite showing the model's stability under certain conditions, such as mixed cell types, unknown cell types, partial marker gene information, and contaminated information, these analyses are limited in scope as they are primarily focused on the human breast tissue marker gene dataset. It is essential to evaluate GPT-4's performance under more diverse scenarios and data types in order to generalize the findings to other tissues, cell types, or disease conditions.

4. While the authors have discussed the cost-effectiveness of their approach from time and the computational effort, they have not provided a quantitative analysis of the actual costs associated with using GPT-4, such as API usage fees. This financial cost assessment is crucial for potential users when deciding whether to adopt this approach.

5. The GPTCelltype R package currently allows users to pass the OpenAI API key directly as a function parameter, posing a risk of inadvertently exposing the key to unauthorized users or accidentally committing it to a version control system. It can be improved by implementing alternative approaches for handling the API key, such as encouraging users to set the OpenAI API key as an environment variable or providing a setup or initialization function for configuring the API key before running primary functions.

Reviewer #2:

Remarks to the Author:

The authors have effectively addressed the major concerns previously raised. Their release of the GPTCelltype software significantly streamlines their proposed cell type annotation process, an admirable contribution.

A minor point to consider: the manuscript might benefit from a brief discussion on the "cost-effectiveness" aspect. In addition to discussing the running time, it is important to highlight that GPT-4 is not free and comes with a monthly subscription fee of \$20, while other methods are free.

Author Rebuttal, first revision:

Response to Reviewers

We would like to thank the editors and reviewers for the valuable feedback to further enhance our manuscript. We appreciate that both reviewers recognize our efforts in addressing the concerns in the

previous round of revision. In this revision, we have carefully addressed reviewers' comments and provided detailed responses below. Changes in the manuscript have been marked in blue color.

Reviewers' Comments:

Reviewer #1:

Remarks to the Author:

I appreciate the authors' efforts to address some of the concerns raised in the initial review. The revised manuscript has refocused on GPT-4's assessment in cell type annotation tasks from scRNA-seq, improving the methodology, developing the GPTCelltype R package, and providing more analyses. However, I still have the following concerns:

1. The authors have not thoroughly assessed various prompt engineering strategies, which are crucial for utilizing large language models like GPT-4. These strategies can significantly influence the model's performance, and it is currently unclear whether the observed improvements in GPT-4's performance are due to the model itself or the specific prompt engineering strategy employed. A systematic assessment of different prompt engineering strategies, including comparisons of prompt formulations, data preprocessing approaches, and input/output encoding methods, is needed to clarify the factors contributing to the performance improvement. Moreover, the authors should provide a detailed discussion of the prompt engineering strategies tested and the rationale behind their final choice to help readers understand the impact of prompt engineering on GPT-4's performance in cell type annotation and make informed decisions when employing GPTCelltype or similar approaches in their own research. Without a systematic assessment of prompt engineering strategies, it remains unclear whether the reported performance improvements are a direct result of GPT-4's capabilities or an artifact of the specific prompt formulation employed in this study.

We thank the reviewer for raising this point. We acknowledge that in our previous revisions, different prompt engineering strategies were not systematically compared. In this revision, we have provided a systematic comparison across several prompt engineering strategies in all applicable datasets. Note that there are infinitely many ways to design prompts and many options in the computational pipeline to obtain differential genes as inputs to GPT-4. As a result, comprehensively evaluating all of them is unrealistic, especially when the evaluation process requires the manual comparison to CL ontology terms and human annotations, which requires a lot of time and effort. Thus, we only demonstrated the impact of some example prompt engineering strategies in this study.

- **Comparisons of different prompt strategy**

To test the impact of prompt formulation, we have included two popular prompt formulations based on chain-of-thought (Wei, J. et al. 2022) and repeated prompt in addition to the existing prompt formulation. In the chain-of-thought prompt formulation, we added intermediate natural language reasoning steps to the existing prompt message. Specifically, the following sentence was added to the basic prompt message: “Because CD3 gene is a marker gene of T cells, if CD3 gene is included in the marker gene list of an unknown cell type, the cell type is likely to be T cells, a subtype of T cells, or a mixed cell type containing T cells.” In the repeated prompt formulation, the basic prompt was fed to GPT-4 five times, and the term that appears most frequently was captured as the final cell type annotation term. We tested the performance of the three prompting strategies (including the basic prompting strategy) in all datasets. The results, which are shown in the newly added Figure 2a, suggest that the performances of different prompting strategies are similar, while the basic prompting strategy has a slightly better performance. A potential reason is that the cell type annotation process is a relatively simple process that does not involve complicated reasoning. Thus, the chain-of-thought prompting strategy may not help much. In addition, results generated by GPT-4 are highly reproducible, which is demonstrated in the reproducibility analysis. Thus, the repeated prompt strategy does not help much either. For simplicity, we decided to use basic prompting strategy as the default prompting strategy throughout the study and in the GPTCelltype R software package.

In addition to the newly added Figure 2a, we added the following detailed discussions of prompting strategies tested to the main section of the manuscript:

“The performance of GPT-4 can also be affected by how the input prompt message was structured. We tested a basic prompt strategy that only includes the necessary information, a prompt strategy inspired by chain-of-thought (CoT) that includes intermediate reasoning steps of an example, and a repeated prompt strategy where GPT-4 was queried multiple times and the most frequently appearing term was selected (Methods, Supplementary Table 3). Results show that the performances of different prompt strategies are comparable (Figure 2a). A potential reason is that cell type annotation does not involve complicated reasoning steps and the results from GPT-4 are highly reproducible. For simplicity, we used the basic prompt strategy for GPT-4 and GPT-3.5 in subsequent analysis.”

We also added the following descriptions of prompting strategies to the methods section of the manuscript:

“For chain-of-thought (CoT) prompt strategy, the following sentence was added to the beginning of the message generated by the basic prompt strategy: “Because CD3 gene is a marker gene of T cells, if CD3 gene is included in the marker gene list of an unknown cell type, the cell type is likely to be T cells, a subtype of T cells, or a mixed cell type containing T cells.””

“For repeated prompt strategy, GPT-4 was queried with the basic prompt strategy repeatedly for five times. The annotation result that appears most frequently among the five queries was selected as the final cell type annotation.”

- **Comparisons of differential analysis methods**

We then evaluated how the choice of statistical method for performing differential analysis would affect the performance of cell type annotation. We choose to evaluate differential gene methods instead of data preprocessing methods since differential gene analysis is the last step of the whole pipeline for identifying differential genes and has the most direct impact to the results. We evaluated the performance of GPT-4 using differential genes obtained from the Wilcoxon rank-sum test or from two-sample t -tests. The results, which are shown in the newly added Figure 2a, suggest that the results from the two differential methods are similar, while Wilcoxon rank-sum test has a slightly better performance. A potential reason is that the Wilcoxon test is more robust and generates differential genes that better represent the identity of the cell type. Since the Wilcoxon test has better performance, we used the Wilcoxon test in the subsequent analysis of the study. In addition, we have also demonstrated the impact of the number of top differential genes on the cell type annotation performance in the previous round of revision.

In addition to the newly added Figure 2a, we added the following contents to the main section of the manuscript:

“We then evaluated the impact of different statistical methods for differential analysis on the performance of cell type annotation (Supplementary Table 3). The annotation performance marginally improves when differential genes are derived using the two-sided Wilcoxon test, as opposed to those obtained through the two-sided two-sample t -test (Figure 2a). This improvement could be attributed to the two-sided Wilcoxon test's robustness, leading to differential genes more aligned with the specific cell type. In subsequent analyses, marker genes provided by Azimuth and literature search datasets and top 10 differential genes obtained from two-sided Wilcoxon test in all other datasets were used as inputs for GPT-4, GPT-3.5, and CellMarker2.0. SingleR and ScType were not performed on Azimuth and literature search datasets since full gene expression matrices were not available.”

We also added the following contents to the methods section of the manuscript:

“In addition, two-sided two-sample t -test was also performed for differential analysis using the FindAllMarkers() function with default settings.”

- **Comparisons of input and output encoding method**

We would like to argue that as an average user, we are unable to adjust the input and output encoding method of GPT-4's Transformer architecture, which can only be done by the OpenAI company. The GPT-4 model itself is not open source, and it is not possible to download the model from some online sources and adjust the architecture by ourselves. The only thing we can do is to explore different ways of prompting strategies, which we have already discussed in detail above and in the manuscript.

In summary, our newly added analyses have shown that the performance of GPT-4 for cell type annotation is quite robust in terms of different prompting strategies and different statistical methods used for identifying differential genes.

Reference:

Wei, J., Wang, X., Schuurmans, D., Bosma, M., Xia, F., Chi, E., Le, Q.V. and Zhou, D., 2022. Chain-of-thought prompting elicits reasoning in large language models. *Advances in Neural Information Processing Systems*, 35, pp.24824-24837.

2. While the authors' new analysis provides a more comprehensive assessment of GPT-4's performance in various challenging scenarios, it does not directly address the concern of overfitting. Overfitting occurs when a model adapts too well to the training data and fails to perform well on unseen data. The simulation studies described in the revised manuscript indeed evaluate GPT-4's performance in differentiating mixed and single cell types, distinguishing known and unknown cell types, and handling partial or contaminated marker gene information. However, these analyses do not directly address the concern of overfitting and the generalizability of GPT-4's performance to new, unseen data. The analysis on known and unknown cell types does not provide a direct measure of GPT-4's performance on new data that could include cell types from different species, tissues, or experimental conditions not seen during the training phase.

We thank the reviewer for this suggestion. In this revision, we have included a new scRNA-seq dataset (Chen, D. et al. 2021) that includes cell type annotations in five tissues from three non-model mammals (cat, tiger, and pangolin). The dataset was published after September 2021, thus not included in the training data

of GPT-4. As stated in the paper (Chen, D. et al. 2021), the scRNA-seq datasets of non-model species are lacking before the publication of Chen, D. et al. 2021. Thus, this dataset includes cell types from species not seen during the training phase of GPT-4. GPT-4 still shows superior performance in this dataset (Figure 2b) and again outperforms other competing methods (Figure 2d), suggesting that overfitting is not a major concern for GPT-4.

This new dataset, as well as the four real datasets that have already been included in the previous revision of the manuscript (Tabula Sapiens, GTEx, literature search (GTEx), and B cell lymphoma), are all datasets published after September 2021 which were not seen in the training phase of GPT-4. In addition, the simulation datasets create cell type marker genes that are randomly generated and unlikely to be seen in the training data. GPT-4 has demonstrated reliable performances in all these datasets and outperforms existing cell type annotation methods. Given this evidence, we believe that overfitting should not be a major concern for GPT-4.

We have added results of the new non-model mammal dataset to all applicable figures, including Figure 1d, 2a, 2b, 2d, and 2f.

Reference:

Chen, D., Sun, J., Zhu, J., Ding, X., Lan, T., Wang, X., Wu, W., Ou, Z., Zhu, L., Ding, P. and Wang, H., 2021. Single cell atlas for 11 non-model mammals, reptiles and birds. *Nature Communications*, 12(1), p.7083.

3. The authors have made efforts to evaluate the influence of input data size and quality on GPT-4's performance by conducting simulation studies on human breast cancer data. Despite showing the model's stability under certain conditions, such as mixed cell types, unknown cell types, partial marker gene information, and contaminated information, these analyses are limited in scope as they are primarily focused on the human breast tissue marker gene dataset. It is essential to evaluate GPT-4's performance under more diverse scenarios and data types in order to generalize the findings to other tissues, cell types, or disease conditions.

We thank the reviewer for this suggestion. In this revision, we have repeated all simulation studies and reproducibility analysis in two additional settings of human colon cancer and human vasculature tissue of the TS dataset, in addition to human breast from GTEx literature search that is already included in the previous revision. Now the simulation studies are done in three tissue types, in both normal and cancer conditions, in three different studies, and using genes that are both from literature search and from differential analysis.

The results, as shown in updated Figure 2g and Figure 2h, suggest that the performance of GPT-4 is comparable to that in the previous revision where only human breast was included. Specifically, GPT-4 is able to (a) differentiate between mixed and homogeneous cell types, (b) differentiate between known and unknown cell types, (c) retains reliable performances with increased levels of noise or reduced amount of information, and (d) demonstrates high level of reproducibility.

These new simulation results, as well as the systematic evaluations of GPT-4's performance in ten real datasets, have demonstrated GPT-4's performance in diverse scenarios and data types and suggest that the findings can be generalized to various tissues, cell types, and disease conditions.

In addition to the updated Figure 2g and Figure 2h, we have added the following descriptions to the methods section of the revised manuscript:

“To generate simulation datasets, we used canonical cell type markers through GTEx literature search of human breast cells, top 10 differential genes from the human colon cancer dataset, and top 10 differential genes from the vasculature tissue of the TS dataset as templates. Simulation studies were performed separately for each of the three tissue types.”

4. While the authors have discussed the cost-effectiveness of their approach from time and the computational effort, they have not provided a quantitative analysis of the actual costs associated with using GPT-4, such as API usage fees. This financial cost assessment is crucial for potential users when deciding whether to adopt this approach.

We thank the reviewer for raising this point. We have provided an assessment of the financial cost of GPT-4 including the monthly subscription fee and an estimate of API usage fee for each cell type annotation query conducted through `GPTCelltype` R package.

We have added new Figure 2f to display how the financial cost of using GPT-4 API changes with the number of cell types in the query. We have added the following descriptions to the main section of the manuscript.

“Compared to other methods that are free of charge, GPT-4 requires a \$20 monthly subscription fee for using the online web portal. The financial cost for using GPT-4 API is linearly correlated with the number of cell types in the query, and does not exceed \$0.1 for all queries in this study (Figure 2f).”

We have also added the below descriptions to the methods section of the manuscript:

GPT-4 API financial cost: According to information provided by OpenAI, the API cost for running GPT-4 June 13, 2023 version is \$0.03 for every thousand input tokens and \$0.06 for every thousand output tokens. For each query, we obtained i and o , which represent the numbers of input tokens and output tokens respectively, through the OpenAI API. The total API financial cost is thus calculated as $\$(0.00003i+0.00006o)$.”

5. The GPTCelltype R package currently allows users to pass the OpenAI API key directly as a function parameter, posing a risk of inadvertently exposing the key to unauthorized users or accidentally committing it to a version control system. It can be improved by implementing alternative approaches for handling the API key, such as encouraging users to set the OpenAI API key as an environment variable or providing a setup or initialization function for configuring the API key before running primary functions.

We thank the reviewer for raising this point. We have revised the GPTCelltype R package so that the OpenAI API key is set as a system environment variable. The value of the API key is provided by the user in an initialization process to avoid the risk of exposing the key. This is achieved by applying `“Sys.setenv(OPENAI_API_KEY = ‘user_openai_api_key’)”` to set the API key before running the functions in the GPTCelltype R package. In the R function `gptcelltype`, we inserted a line of code `“OPENAI_API_KEY <- Sys.getenv(“OPENAI_API_KEY”)”` to retrieve the API key values set by the user.

These procedures are detailed in the vignette page and user manual of the software. We have added a new section called “Set up OpenAI API key as an environment variable” in the vignette page and the user manual. Specifically, we have marked that “To avoid the risk of exposing the API key or committing the key to browsers, users need to set up the API key as a system environment variable before running GPTCelltype.” and “Set up the API key as a system environment variable before running GPTCelltype.

```
Sys.setenv(OPENAI_API_KEY = 'your_openai_API_key')
```

The complete updated vignette can be accessed through https://winnie09.github.io/Wenpin_Hou/pages/gptcelltype.html

Reviewer #2:

Remarks to the Author:

The authors have effectively addressed the major concerns previously raised. Their release of the GPTCelltype software significantly streamlines their proposed cell type annotation process, an admirable contribution.

A minor point to consider: the manuscript might benefit from a brief discussion on the "cost-effectiveness" aspect. In addition to discussing the running time, it is important to highlight that GPT-4 is not free and comes with a monthly subscription fee of \$20, while other methods are free.

We thank the reviewer for raising this point. We have provided an assessment of the financial cost of GPT-4 including the monthly subscription fee and an estimate of API usage fee for each cell type annotation query conducted through GPTCelltype R package.

We have added new Figure 2f to display how the financial cost of using GPT-4 API changes with the number of cell types in the query. We have added the following descriptions to the main section of the manuscript.

“Compared to other methods that are free of charge, GPT-4 requires a \$20 monthly subscription fee for using the online web portal. The financial cost for using GPT-4 API is linearly correlated with the number of cell types in the query, and does not exceed \$0.1 for all queries in this study (Figure 2f).”

We have also added the below descriptions to the methods section of the manuscript:

“**GPT-4 API financial cost:** According to information provided by OpenAI, the API cost for running GPT-4 June 13, 2023 version is \$0.03 for every thousand input tokens and \$0.06 for every thousand output tokens. For each query, we obtained i and o , which represent the numbers of input tokens and output tokens respectively, through the OpenAI API. The total API financial cost is thus calculated as $\$(0.00003i+0.00006o)$.”

Decision Letter, second revision:

Our ref: NMETH-BC52314B

13th Dec 2023

Dear Dr. Ji,

Thank you for submitting your revised manuscript "Assessing GPT-4 for cell type annotation in single-cell RNA-seq analysis" (NMETH-BC52314B). It has now been seen by the original referees and their comments are below. The reviewers find that the paper has improved in revision, and therefore we'll be happy in principle to publish it in Nature Methods, pending minor revisions to comply with our editorial and formatting guidelines.

TRANSPARENT PEER REVIEW

ORCID

Sincerely,

Lin Tang, PhD
Senior Editor
Nature Methods

Reviewer #1 (Remarks to the Author):

The authors have addressed my concerns

Final Decision Letter:

5th Mar 2024

Dear Dr Ji,

I am pleased to inform you that your Brief Communication, "Assessing GPT-4 for cell type annotation in single-cell RNA-seq analysis", has now been accepted for publication in *Nature Methods*. The received and accepted dates will be 16th Apr 2023 and 5th Mar 2024. This note is intended to let you know what to expect from us over the next month or so, and to let you know where to address any further questions.

Over the next few weeks, your paper will be copyedited to ensure that it conforms to *Nature Methods* style. Once your paper is typeset, you will receive an email with a link to choose the appropriate publishing options for your paper and our Author Services team will be in touch regarding any additional information that may be required.

Once proofs are generated, they will be sent to you electronically and you will be asked to send a corrected version within 48 hours. It is extremely important that you let us know now whether you will be difficult to contact over the next month. If this is the case, we ask that you send us the contact information (email, phone and fax) of someone who will be able to check the proofs and deal with any last-minute problems.

If, when you receive your proof, you cannot meet the deadline, please inform us at rjsproduction@springernature.com immediately.

Please note that *Nature Methods* is a Transformative Journal (TJ). Authors may publish their research with us through the traditional subscription access route or make their paper immediately open access

through payment of an article-processing charge (APC). Authors will not be required to make a final decision about access to their article until it has been accepted. Find out more about Transformative Journals

To assist our authors in disseminating their research to the broader community, our SharedIt initiative provides you with a unique shareable link that will allow anyone (with or without a subscription) to read the published article. Recipients of the link with a subscription will also be able to download and print the PDF. As soon as your article is published, you will receive an automated email with your shareable link.

Please note that you and your coauthors may order reprints and single copies of the issue containing your article through Springer Nature Limited's reprint website, which is located at <http://www.nature.com/reprints/author-reprints.html>. If there are any questions about reprints please send an email to author-reprints@nature.com and someone will assist you.

Please feel free to contact me if you have questions about any of these points. Thank you very much for publishing your work at Nature Methods!

Best regards,

Lin Tang, PhD
Senior Editor
Nature Methods